# The human MRS2 magnesium-binding domain is a regulatory feedback switch for channel activity

Sukanthathulse Uthayabalan[1], Neelanjan Vishnu[2], Muniswamy Madesh[2], Peter B Stathopulos[1]

**Mitochondrial RNA splicing 2 (MRS2) forms a magnesium ($Mg^{2+}$) entry protein channel in mitochondria. Whereas MRS2 contains two transmembrane domains constituting a pore on the inner mitochondrial membrane, most of the protein resides within the matrix. Yet, the precise structural and functional role of this obtrusive amino terminal domain (NTD) in human MRS2 is unknown. Here, we show that the MRS2 NTD self-associates into a homodimer, contrasting the pentameric assembly of CorA, an orthologous bacterial channel. $Mg^{2+}$ and calcium suppress lower and higher order oligomerization of MRS2 NTD, whereas cobalt has no effect on the NTD but disassembles full-length MRS2. Mutating-pinpointed residues-mediating $Mg^{2+}$ binding to the NTD not only selectively decreases $Mg^{2+}$-binding affinity ~sevenfold but also abrogates $Mg^{2+}$ binding–induced secondary, tertiary, and quaternary structure changes. Disruption of NTD $Mg^{2+}$ binding strikingly potentiates mitochondrial $Mg^{2+}$ uptake in WT and Mrs2 knockout cells. Our work exposes a mechanism for human MRS2 autoregulation by negative feedback from the NTD and identifies a novel gain of function mutant with broad applicability to future $Mg^{2+}$ signaling research.**

## Introduction

Magnesium ions ($Mg^{2+}$) are the most abundant divalent cations in eukaryotes, playing universal roles in myriad cell functions (Jahnen-Dechent & Ketteler, 2012). Within mitochondria, $Mg^{2+}$ is an important protein-stabilizing cofactor, forms biologically functional $Mg^{2+}$–ATP complexes, and regulates crucial enzymatic activities. Such roles are achieved through two unique properties of $Mg^{2+}$: (i) the ability to form chelates with important intracellular anionic-ligands (i.e., small molecule or large biomolecule), and (ii) the capability to compete with calcium ions ($Ca^{2+}$) for binding sites on proteins and membranes (Berridge et al, 2000; Moomaw & Maguire, 2008; Chaigne-Delalande et al, 2013). The effects of $Mg^{2+}$ on $Ca^{2+}$-handling proteins significantly influence intracellular $Ca^{2+}$

dynamics and signaling (Gregan et al, 2001; Clapham, 2007; de Baaij et al, 2015).

Total cellular $Mg^{2+}$ concentrations range between ~17 and 30 mM; however, concentrations of free $Mg^{2+}$ in the cytosol are estimated between ~0.5 and 1.5 mM (Jung et al, 1990; Rutter et al, 1990; Romani, 2011). Intracellular $Mg^{2+}$ concentrations are strongly buffered and regulated by the combined action of $Mg^{2+}$-binding molecules, $Mg^{2+}$ storage in organelles, and the action of $Mg^{2+}$ channels and exchangers. Remarkably, $Mg^{2+}$ can be mobilized from the ER in response to ligands such as L-lactate, moving into the mitochondria and dramatically modifying metabolism (Daw et al, 2020). $Mg^{2+}$ can also alter the electrophysiological properties of ion channels such as voltage-dependent $Ca^{2+}$ channels and potassium ($K^+$) channels and affect the binding affinity of $Ca^{2+}$ to EF-hand–containing proteins (Glancy & Balaban, 2012; Pilchova et al, 2017). All ATP-related biochemical reactions in cells are dependent on $Mg^{2+}$ (Romani, 2007), and extracellular $Mg^{2+}$ also regulates numerous channels such as glutamate receptors and N-methyl-D-aspartate receptors (Kumar, 2015). In addition, this divalent cation contributes to the maintenance of genome stability as a cofactor in DNA repair and protection (Hartwig, 2001).

Unsurprisingly, perturbations in intracellular $Mg^{2+}$ concentrations can cause serious cellular dysfunction. For example, decreases in intracellular free $Mg^{2+}$ lead to defective immune responses (Zhou & Clapham, 2009; Chaigne-Delalande et al, 2013; Kanellopoulou et al, 2019), mutations in the $Na^+/Mg^{2+}$ exchanger causing chronic intracellular $Mg^{2+}$ deficiency trigger neuronal damage (Kolisek et al, 2013), and overexpression of $Mg^{2+}$ channels is a hallmark of several types of cancer (Trapani & Wolf, 2019), to name a few. More specifically in terms of neuronal disease, the A350V mutation in SLC41A1 enhances Na+-dependent $Mg^{2+}$ efflux by ~70% in HEK293 cells (Kolisek et al, 2013), and low $Mg^{2+}$ intake in rats and humans leads to loss of dopaminergic neurons (Oyanagi et al, 2006) and increased risk of idiopathic Parkinson's disease, respectively (Aden et al, 2011). Indeed, decreased free cytosolic $Mg^{2+}$ has been measured in the occipital lobes of Parkinson's patients (Barbiroli et al, 1999). Transient receptor potential melastatin-6 mutations can also be pathophysiological, resulting in $Mg^{2+}$ malabsorption and renal wasting, where

[1]Department of Physiology and Pharmacology, Schulich School of Medicine and Dentistry, University of Western Ontario, London, Canada   [2]Center for Mitochondrial Medicine, Department of Medicine, University of Texas Health San Antonio, San Antonio, TX, USA

Correspondence: pstatho@uwo.ca

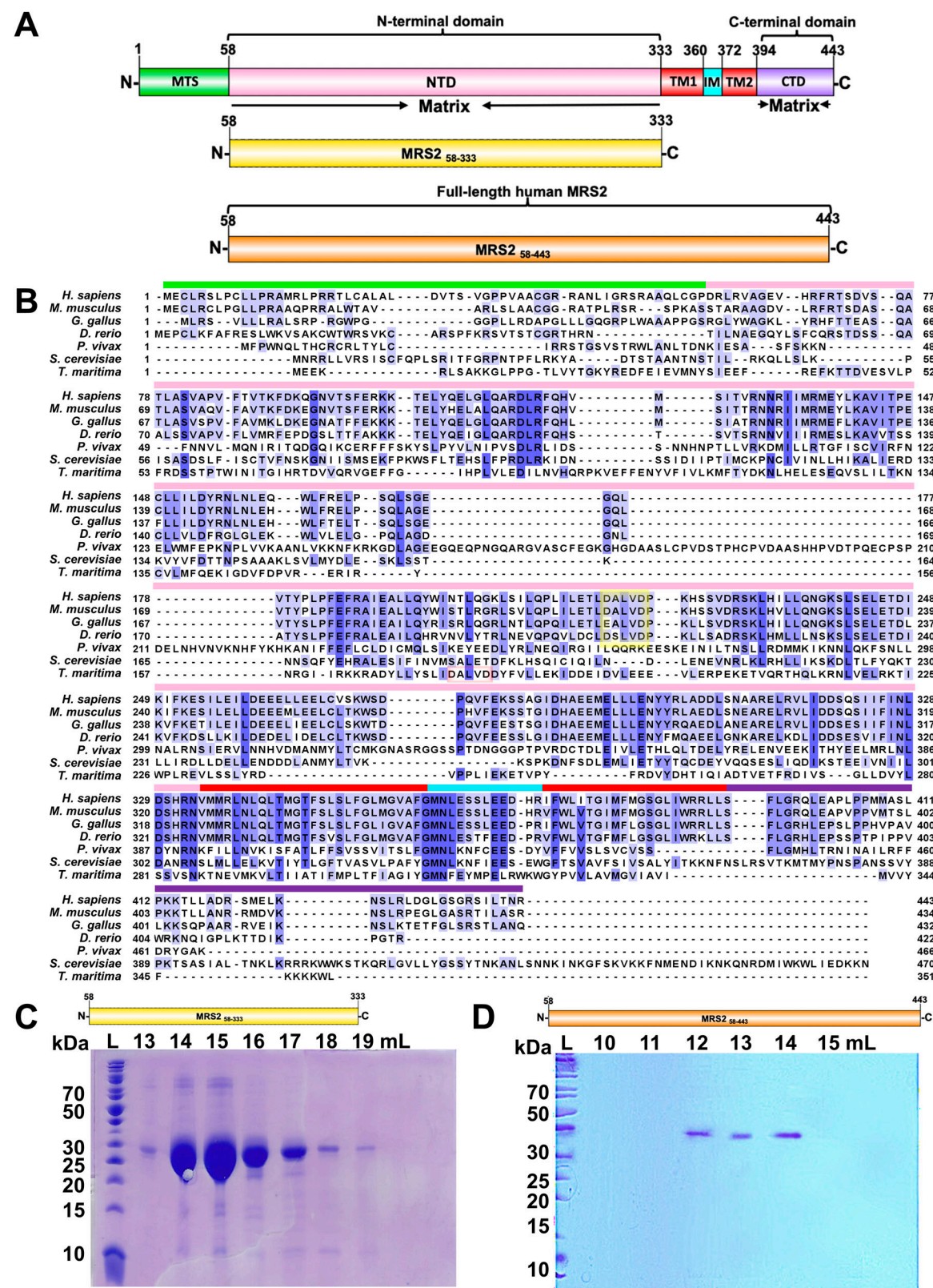

**Figure 1. Domain architecture and sequence alignments of MRS2 family proteins.**
**(A)** Domain architecture of human MRS2. The relative locations of the mitochondrial targeting sequence (MTS, green), amino terminal domain (NTD, magenta), transmembrane 1, 2 (TM1/2, red), intermembrane space (IM, cyan), and C-terminal domain (CTD, violet) are shown. The residue ranges are shown above and below each domain, labeled based on UniProt Q9HD23 and bioinformatics annotations (see the Results section). Below, the N-terminal domain (NTD, yellow) and full-length (orange)

affected individuals show seizures and muscle spasms during infancy (Schlingmann et al, 2002; Schaffers et al, 2018).

Mitochondria have an inner mitochondrial membrane (IMM), separating the mitochondrial matrix (MM) from the intermembrane space, and an outer mitochondrial membrane, enclosing the entire organelle. Unregulated, the highly negative IMM potential (~–180 mV) (Marchi & Pinton, 2014) would drive catastrophically high concentrations of $Mg^{2+}$ entry into the matrix; nevertheless, the matrix $Mg^{2+}$ concentration is similar to the concentration in the cytoplasm, reinforcing that $Mg^{2+}$ influx into the organelle is tightly controlled to maintain optimal mitochondrial function and bioenergetics (Marchi & Pinton, 2014; Pilchova et al, 2017).

Residing within the IMM in mammalian cells, mitochondrial RNA splicing 2 (MRS2) constitutes a major $Mg^{2+}$ entry protein channel into the MM. Deletion of IMM-localized MRS2 abolishes $Mg^{2+}$ influx into the MM, inducing functional defects in mitochondria and promoting cell death (Piskacek et al, 2009; Merolle et al, 2018). MRS2 belongs to the large heterogeneous CorA/Mrs2/Alr1 protein superfamily of $Mg^{2+}$ transporters. This family is characterized by the highly conserved Gly-Met-Asn (GMN) motif at the end of the first transmembrane helix, essential for $Mg^{2+}$ transporter function. Mutations of the GMN motif either completely abolish $Mg^{2+}$ transport or profoundly change the ion selectivity of the channel (Knoop et al, 2005; Palombo et al, 2013; Merolle et al, 2018). Human MRS2 contains a large, amino terminal domain (NTD) oriented within the MM, corresponding to residues 58–333 and consisting of ~71% of the mature polypeptide chain, two transmembrane (TM1 and TM2) domains connected by a highly conserved intermembrane space loop and a smaller, carboxyl terminal domain also oriented within the MM (Fig 1A).

Although CorA and Mrs2, orthologues of MRS2 in bacteria and yeast, respectively, have been structurally resolved at high resolution (Eshaghi et al, 2006; Lunin et al, 2006; Payandeh & Pai, 2006; Guskov et al, 2012; Pfoh et al, 2012; Khan et al, 2013; Matthies et al, 2016; Johansen et al, 2022), low sequence similarity exists between CorA, Mrs2, and MRS2, especially in the NTDs (Zsurka et al, 2001). Specifically, the sequence similarity between human MRS2 and yeast (*Saccharomyces cerevisiae*) Mrs2 is 55.4% (and only 20.1% through the NTD), whereas the sequence similarity between human MRS2 and bacterial (*Thermotoga maritima*) CorA is 43.3% (and only 17.0% through the NTD) (Fig 1B). To reveal how the prominent MRS2 NTD governs the assembly and function of the full channel, we generated recombinant human NTD protein (MRS2$_{58-333}$; residues 58–333) and full-length human MRS2 (MRS2$_{58-443}$; residues 58–443). Using light scattering and chromatographic approaches, we find that in the absence of divalent cations, the NTD self-associates into a homodimer under dilute conditions, whereas both $Mg^{2+}$ and $Ca^{2+}$, but not cobalt ($Co^{2+}$), suppress the self-association of the domain. In contrast, $Co^{2+}$ disassembles full-length MRS2, whereas $Mg^{2+}$ and

$Ca^{2+}$ have no effect on stoichiometry. 8-Anilino-1-naphthalene sulfonate (ANS) and intrinsic fluorescence measurements suggest that $Mg^{2+}$ and $Ca^{2+}$ bind to distinct sites on the NTD with ~$\mu M$ and mM affinity, respectively. Importantly, we identify the D216 and D220 as critical residues for $Mg^{2+}$ coordination to the human MRS2 NTD, where mutating these residues decreases $Mg^{2+}$ affinity ~sevenfold, abrogates $Mg^{2+}$ binding–induced increases in $\alpha$-helicity and solvent accessible hydrophobicity, and suppresses the $Mg^{2+}$-induced disassembly of the NTD. Finally, using both permeabilized and intact cell models, we show that $Mg^{2+}$ binding to the NTD suppresses the rate of $Mg^{2+}$ uptake into mitochondria as negative feedback. Collectively, our data reveal previously unknown mechanistic insights underlying human MRS2 autoregulation by the large NTD, which has important implications for understanding the crosstalk between MM $Mg^{2+}$ concentrations, bioenergetics, and cell death.

# Results

## MRS2 NTD homodimer assembly is sensitive to divalent cations

Given that human *MRS2* encodes only two putative TMs (Fig 1A) and must oligomerize to form a channel pore, we first evaluated the stoichiometry of the MRS2 NTD (i.e., MRS2$_{58-333}$) using size exclusion chromatography with in-line multi-angle light scattering (SEC-MALS). Recombinant MRS2$_{58-333}$ was successfully expressed and isolated with high yield (i.e., ~8 g/l of culture) and purity (Fig 1C). Although the theoretical monomeric molecular weight of MRS2$_{58-333}$ is ~32.2 kD, SEC-MALS revealed that, in the absence of divalent cations, MRS2$_{58-333}$ consistently self-associates into a homodimer with estimated molecular weights of 60.9 ± 1.8 kD and 61.3 ± 0.50 kD at 2.5 and 5.0 mg/ml, respectively (Fig 2A and B). Homodimer formation is apparently tight as single elution peaks and no protein concentration dependence on the molecular weight or elution volume in the 0.45–5 mg/ml range was observed (Table S1).

SEC-MALS was further used to assess the sensitivity of the homodimer assembly to $Mg^{2+}$ and $Ca^{2+}$ because earlier studies showed divalent cation binding to the CorA NTD regulates channel structure and function (Pfoh et al, 2012; Khan et al, 2013) (see also Discussion section). The presence of 5 mM $MgCl_2$ in the elution buffer transitioned the molecular weight of MRS2$_{58-333}$ to monomer at 2.5 mg/ml, with a SEC-MALS molecular weight of 29.4 ± 6.0 kD (Fig 2C). In contrast, MRS2$_{58-333}$ remained dimeric at 5 mg/ml in the presence of 5 mM $MgCl_2$, with a molecular weight of 60.06 ± 0.47 kD (Fig 2D). Adding 10 mM $MgCl_2$ to the 5 mg/ml sample, however, resulted in a monomeric molecular weight of 32.6 ± 0.5 kD (Fig 2E). Remarkably, adding 5 mM $CaCl_2$ to both the 2.5 and 5.0 mg/ml MRS2$_{58-33}$ samples robustly caused monomer formation, with measured molecular weights of 31.4 ± 1.0 and 29.0 ± 0.44

constructs engineered in the present study are shown relative to the entire pre-protein. **(B)** Multiple sequence alignment of MRS2 orthologues. Sequences for human (UniProt accession Q9HD23), *Thermotoga maritima* (Q9WZ31), *Saccharomyces cerevisiae* (Q01926), *Plasmodium Vivax* (A0A1G4H438), *Danio rerio* (E7F680), *Gallus gallus* (A0A1D5P665), and *Mus musculus* (Q5NCE8) were aligned using Clustal Omega with defaults (Sievers et al, 2011) and annotated in Jalview (Waterhouse et al, 2009). **(A)** Coloured bars above the human MRS2 sequence mark the boundaries as per the pre-protein in (A). Yellow and red boxes highlight the location of the DALVD sequence in vertebrates and *T. maritima*, respectively. Residue numbers are shown at left and right of each entry and blue shades correspond to conserved positions. Note that human MRS2 and bacterial CorA alignment is algorithm-dependent due to poor sequence conservation (see Figs S1 and S2). **(C)** Coomassie blue–stained SDS–PAGE gel showing the elution fractions containing human MRS2$_{58-333}$. **(D)** Coomassie blue–stained SDS–PAGE gel showing the elution fractions containing MRS2$_{58-443}$. In (C, D), elution volumes through an S200 10/300 Gl SEC column are shown at top and ladder "L" bands are shown at left.

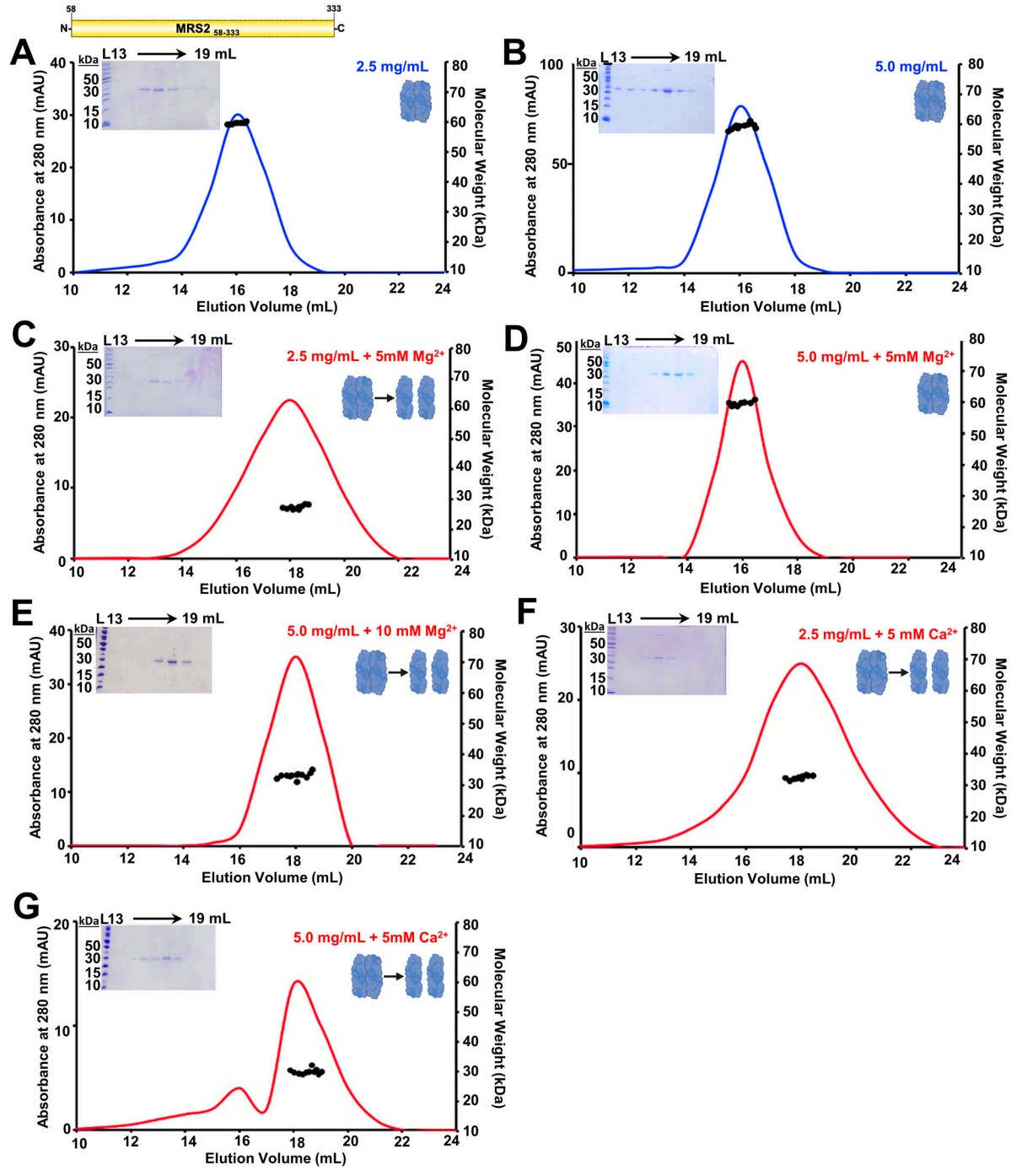

**Figure 2. Quaternary structure of MRS2_{58–333} (NTD).**
**(A, B)** SEC-MALS analysis of MRS2_{58–333} injected at (A) 2.5 mg/ml and (B) 5.0 mg/ml in the absence of divalent cations. **(C, D)** SEC-MALS data of MRS2_{58–333} injected at (C) 2.5 mg/ml and (D) 5.0 mg/ml in the presence of 5 mM MgCl_2. **(E)** SEC-MALS data of MRS2_{58–333} injected at (E) 5.0 mg/ml MRS2_{58–333} in the presence of 10 mM MgCl_2. **(F, G)** SEC-MALS data of MRS2_{58–333} injected at (F) 2.5 mg/ml and (G) 5.0 mg/ml in the presence of 5 mM CaCl_2. In (A, B, C, D, E, F, G), MALS-determined molecular weights are shown through the elution peaks (black circles), left insets show Coomassie blue–stained SDS–PAGE gels of the elution fractions from the 5.0 mg/ml injections and right insets depict the dimerization state of the protein and divalent cation-free and supplemented chromatograms are coloured blue and red, respectively. Elution volumes are indicated at top and ladder (L) molecular weights at left of the gels. Data are representative of n = 3 separate injections from three protein preparations (Table S1) and were acquired using an S200 10/300 Gl column in 20 mM Tris, 150 mM NaCl, and 1 mM DTT, pH 8.0, 10°C.

kD, respectively (Fig 2F and G). Supplementation with 5 mM MgCl$_2$ to MRS2$_{58-333}$ samples at less than 2.5 mg/ml was sufficient to cause monomerization, and shifts to later elution volumes were consistent with all divalent cation-dependent disassembly observations (Table S1).

## Divalent cations regulate MRS2 assembly in a domain-specific manner

Because SEC-MALS was performed at 10°C and is accompanied by a large column dilution (i.e., minimum ~20-fold) that could affect assembly, we used dynamic light scattering (DLS) to assess the distribution of hydrodynamic radii (R$_h$) at 1.25 mg/ml in the absence of dilution and at higher temperature (i.e., 20 and 37°C). Bimodal distributions of R$_h$ centered at ~4 and 40 nm were observed for MRS2$_{58-333}$ in the absence of divalent cations at both 20 and 37°C (Fig 3A–F). The addition of either 5 mM CaCl$_2$ (Fig 3A and B) or 5 mM MgCl$_2$ (Fig 3C and D) at both temperatures eliminated the larger size distributions. The loss of larger R$_h$ was supported qualitatively by earlier decays in the autocorrelation functions when compared with divalent cation-free protein samples (Fig 3A–D, insets). The bacterial orthologue of MRS2 and CorA can transport Co$^{2+}$ and Mg$^{2+}$, and Co$^{2+}$ is found in trace levels in mammals (Tapiero et al, 2003; Guskov & Eshaghi, 2012; Czarnek et al, 2015); thus, we also assessed the sensitivity of the MRS2 NTD assembly to Co$^{2+}$. The distribution of R$_h$ was not affected by the addition of 5 mM CoCl$_2$ at either 20 or 37°C (Fig 3E and F).

Next, we evaluated the effects of Mg$^{2+}$, Ca$^{2+}$, and Co$^{2+}$ on the assembly of the full-length protein. Full-length MRS2, excluding the mitochondrial targeting sequence, (MRS2$_{58-443}$), was successfully expressed and isolated with high purity (Fig 1D). The experimental buffer for MRS2$_{58-443}$ included CHAPS and showed autocorrelation functions consistent with the presence of ~1–1.5 nm micelles at 37°C (Fig 4A–C). MRS2$_{58-443}$ samples showed autocorrelation functions with later decay times compared with buffer alone, which were deconvoluted to R$_h$ distributions centered at ~4 and ~20 nm at 37°C (Fig 4A–C). In contrast to the NTD data, changes in autocorrelation functions and size distributions were not observed with MRS2$_{58-443}$ when supplemented with either 5 mM MgCl$_2$ or 5 mM CaCl$_2$ (Fig 4A and B). Remarkably, 5 mM CoCl$_2$ completely abrogated the larger R$_h$ distributions, which was qualitatively supported by autocorrelation function shifts to earlier decay times (Fig 4C).

Applying a cumulative deconvolution to extract one weight-averaged R$_h$ from all autocorrelation functions reinforced the regularization/polydisperse deconvolution trends described above. Specifically, CoCl$_2$ caused a robust decrease in the weight-averaged R$_h$ for full-length MRS2, but not the NTD; MgCl$_2$ and CaCl$_2$ decreased R$_h$ for the NTD, but not full-length MRS2 (Table S2). Collectively, these data suggest a domain-specific sensitivity to divalent cations, where NTD disassembly is promoted by Mg$^{2+}$ and Ca$^{2+}$, whereas Co$^{2+}$ de-oligomerizes full-length MRS2 because of sensitivity outside the NTD.

## Mg$^{2+}$ and Ca$^{2+}$ bind to distinct sites on the MRS2 NTD with disparate affinities

Given that both Mg$^{2+}$ and Ca$^{2+}$ dissociate MRS2$_{58-333}$, which contains 3×Trp and 7×Tyr residues, we next used changes in intrinsic fluorescence

to evaluate divalent cation binding. Fluorescence emission spectra were acquired using an excitation wavelength of 280 nm as a function of increasing MgCl$_2$, CaCl$_2$, and CoCl$_2$ concentrations. The intensities of the fluorescence emission spectra decreased as a function of increasing MgCl$_2$ (Fig 5A) and CaCl$_2$ (Fig 5B) concentrations. Both MgCl$_2$ and CaCl$_2$ effects were saturable; however, the intensity decreased by ~32% with Mg$^{2+}$ and only ~5% with Ca$^{2+}$, suggesting distinct structural effects and/or binding sites. In contrast, titration with CoCl$_2$ caused small increases of ~2% in fluorescence (Fig 5C). Fitting the binding curves to a one-site binding model that accounts for protein concentrations revealed apparent equilibrium dissociation constants (K$_d$)s of ~0.14 ± 0.03, 1.01 ± 0.26, and 0.68 ± 0.30 mM for Mg$^{2+}$, Ca$^{2+}$, and Co$^{2+}$ interactions, respectively (Table S3).

We next attempted to pinpoint the residues involved in Mg$^{2+}$ coordination using the CorA crystal structure (4EED.pdb) (Pfoh et al, 2012) as a guide. Note that available yeast Mrs2 structures do not resolve any Mg$^{2+}$ ions bound to the NTD. The *T. maritima* CorA crystal structure shows that two Asp residues, separated by three residues (i.e., DALVD) are involved in Mg$^{2+}$ coordination at one site. Remarkably, human MRS2 contains the same DALVD sequence stretch in the homologous domain, which we posited could similarly coordinate Mg$^{2+}$ (Fig 1B). However, sequence-based alignment of the bacterial and vertebrate DALVD regions are algorithm-dependent because of the poor sequence conservation: T-Coffee (Notredame et al, 2000) aligns the human and *T. maritima* DALVD stretches as conserved (Fig S1), whereas Clustal Ω (Sievers et al, 2011) does not (Figs 1B and S2). Moreover, the AlphaFold2 prediction for human MRS2 (see the Discussion section) agrees with the non-conserved Clustal Ω positioning of bacterial and vertebrate DALVD regions. Thus, our use of the term DALVD with respect to human MRS2 refers to the stretch of polypeptide chain containing the D216 and D220 residues and does not imply a motif that is conserved with *T. maritima* CorA DALVD. We use the term DALVD to highlight that the two Asp residues are at positions *i* and *i*+4 of this sequence stretch with no intervening helix-breaking residues, putatively orienting these side chains on the same side of an α-helix and adjacent in three-dimensional (3D) space and permitting both Asp to interact with the same Mg$^{2+}$ ion.

After creating a D216A/D220A MRS2$_{58-333}$ double mutant, we reassessed divalent cation binding by intrinsic fluorescence. A double mutant was created because both Asp side chains coordinate the same Mg$^{2+}$ ion in the CorA DALVD sequence, both Asp side chains are close in 3D space in the α-helix where most of the human MRS2 region is predicted to exist by AlphaFold2 (see the Discussion section), and the sub-mM Mg$^{2+}$ K$_d$ (Table S3) suggests both Asp are involved in the coordination. Not only did the D216A/D220A mutant show a small increase in fluorescence (Mg$^{2+}$ causes a large decrease in the fluorescence intensity of WT MRS2$_{58-333}$; see above) but also a markedly suppressed intensity change as a function of increasing MgCl$_2$, consistent with perturbation of Mg$^{2+}$ binding (Fig 5D). In contrast, the CaCl$_2$ and CoCl$_2$ effects were similar to data acquired using WT MRS2$_{58-333}$ (Fig 5E and F). Indeed, fitting the datasets to one-site binding models revealed apparent equilibrium dissociation constants (K$_d$) of ~0.98 ± 0.25, 0.74 ± 0.49, and 1.37 ± 0.51 mM (Table S3), consistent with disruption of the Mg$^{2+}$ interactions but not Ca$^{2+}$ or Co$^{2+}$.

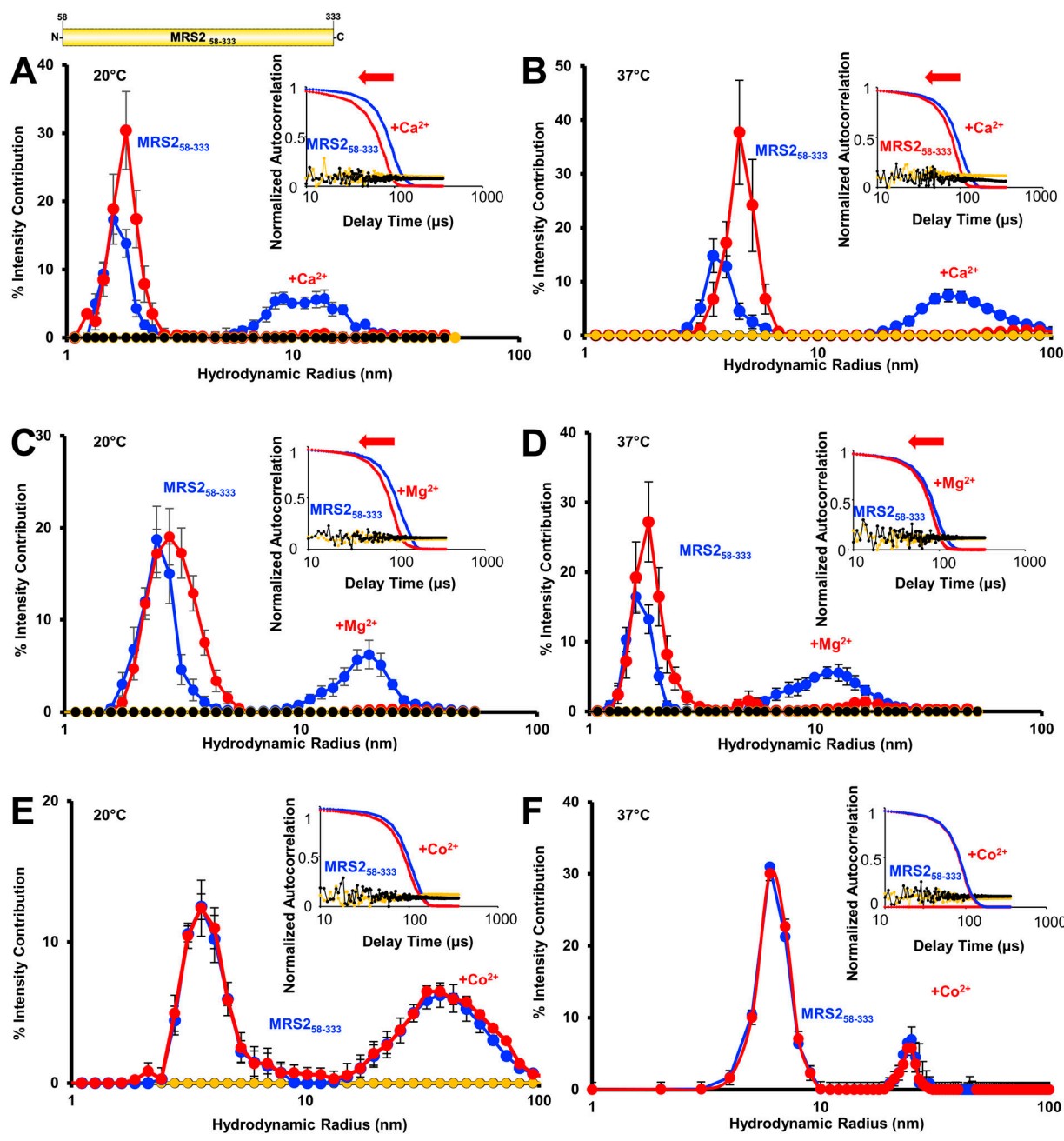

**Figure 3. Higher order oligomerization of MRS2$_{58-333}$ (NTD).**
**(A, B)** Distributions of hydrodynamic radii (R$_h$) from the regularization deconvolution of the autocorrelation functions in the presence and absence of 5 mM CaCl$_2$ at (A) 20°C and (B) 37°C. **(C, D)** Distributions of R$_h$ from the regularization deconvolution of the autocorrelation functions in the presence and absence of 5 mM MgCl$_2$ at (C) 20°C and (D) 37°C. **(E, F)** Distributions of R$_h$ from the regularization deconvolution of the autocorrelation functions in the presence and absence of 5 mM CoCl$_2$ at (E) 20°C and (F) 37°C. In (A, B, C, D, E, F), insets show the divalent cation-induced shifts in the autocorrelation functions, divalent cation-free protein sample data are coloured blue, divalent cation-supplemented protein sample data are red, divalent cation-free buffer control data are black, and divalent cation-supplemented buffer control data are yellow. Inset data are representative, while deconvoluted R$_h$ profiles are means ± SEM of n = 3 separate samples from three protein preparations. All data were acquired at 1.25 mg/ml protein in 20 mM Tris, 150 mM NaCl, and 1 mM DTT, pH 8.0.

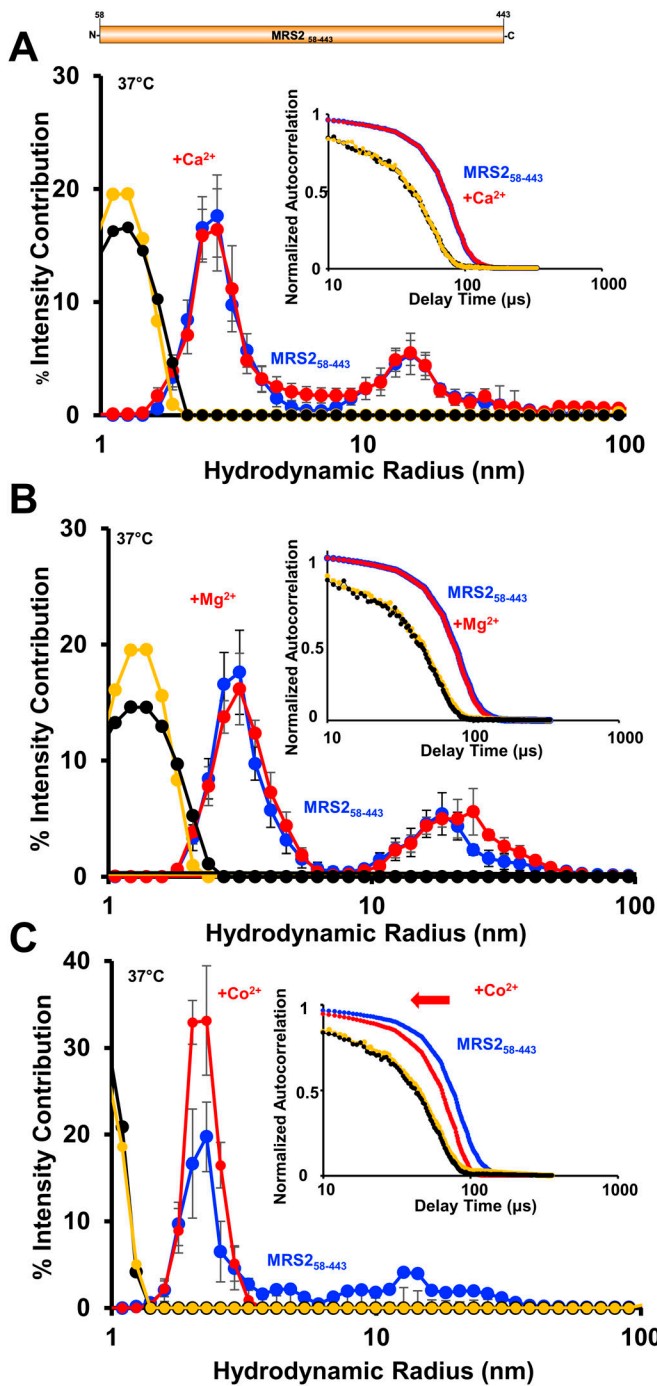

**Figure 4. Higher order oligomerization of MRS2$_{58-443}$ (full-length).**
**(A)** Distributions of R$_h$ from the regularization deconvolution of the autocorrelation functions in the presence and absence of 5 mM CaCl$_2$. **(B)** Distributions of R$_h$ from the regularization deconvolution of the autocorrelation functions in the presence and absence of 5 mM MgCl$_2$. **(C)** Distributions of R$_h$ from the regularization deconvolution of the autocorrelation functions in the presence and absence of 5 mM CoCl$_2$. In (A, B, C), insets show the divalent cation-induced shifts in the autocorrelation functions, divalent cation-free protein sample data are coloured blue, divalent cation-supplemented protein sample data are red, divalent cation-free buffer control data are black and divalent cation-supplemented buffer control data are yellow. Inset data are representative, whereas deconvoluted R$_h$ profiles are means ± SEM of n = 3 separate samples from three protein preparations. All data were acquired at 0.5 mg/ml in 20 mM Tris, 150 mM NaCl, 1 mM DTT, and 10 mM CHAPS, pH 8.0, 37°C.

Taken together, these data suggest that Mg$^{2+}$ and Ca$^{2+}$ bind to distinct sites on the MRS2NTD, with D216 and D220 mediating interactions with Mg$^{2+}$.

## Mg$^{2+}$ enhances whereas Ca$^{2+}$ suppresses solvent exposed hydrophobicity of MRS2 NTD

Given the changes in stoichiometry observed by DLS and SEC-MALS, we next assessed the solvent-exposed hydrophobicity of MRS2$_{58-333}$ in the absence and presence of Mg$^{2+}$ and Ca$^{2+}$ by monitoring extrinsic ANS fluorescence. ANS binds to solvent accessible hydrophobic regions on biomolecules, resulting in a blue-shifted fluorescence emission maximum and increased intensity (Stryer, 1965). Baseline fluorescence emission spectra of ANS in the presence of buffer alone were insensitive to the addition of 5 mM MgCl$_2$ or 5 mM CaCl$_2$ (Fig S3A and B). Indeed, ANS binding was detected in the presence of 2.5 mg/ml MRS2$_{58-333}$, as evidenced by the blue-shifted fluorescence emission maximum and increased intensity compared with the buffer controls (Fig S3A and B). Supplementing the protein samples with 5 mM MgCl$_2$ caused a small but significant increase in ANS fluorescence intensity, suggesting enhanced exposed hydrophobicity (Fig 6A and B). Conversely, supplementation with 5 mM CaCl$_2$ caused a small but significant decrease in ANS fluorescence intensity, indicating decreased solvent exposed hydrophobicity (Fig 6C and D).

We next performed a similar set of experiments with the D216A/D220A MRS2$_{58-333}$ protein. Consistent with our observation that this double mutant disrupts Mg$^{2+}$ but not Ca$^{2+}$ binding to the NTD; ANS emission spectra in the presence of protein showed no differences with or without MgCl$_2$ supplementation (Figs 6E and F and S3C), whereas CaCl$_2$ supplementation caused a small but significant decrease in ANS fluorescence intensity (Figs 6G and H and S3D). This ANS binding data reinforces the notion of disparate Ca$^{2+}$- and Mg$^{2+}$-binding sites and suggests that these divalent cations may cause distinct MRS2 NTD conformational changes.

## D216A/D220A mitigates Mg$^{2+}$-dependent disassembly of the MRS2 NTD

Next, we tested whether the D216A/D220A double mutation could abolish the Mg$^{2+}$-dependent monomerization and decreased R$_h$ observed with WT MRS2 NTD. In the absence of the divalent cation, SEC-MALS revealed that D216A/D220A MRS2$_{58-333}$ elutes as a homodimer with a molecular weight of 59.7 ± 2.0 kD when injected at 2.5 mg/ml (Fig 7A), similar to WT MRS2$_{58-333}$ (Table S1). In contrast to WT evaluated at 2.5 mg/ml, however, the SEC-MALS–determined molecular weight of the double mutant remained dimeric (i.e., 59.0 ± 2.4 kD) after the addition of 5 mM MgCl$_2$ (Fig 7B). We also assessed whether Mg$^{2+}$ could alter R$_h$ of the D216A/D220A MRS2$_{58-333}$ by DLS. Addition of 5 mM MgCl$_2$ neither altered the distribution of R$_h$ nor the autocorrelation function compared with samples evaluated in the absence of the cation (Fig 7C). It is to be noted that a bimodal distribution of R$_h$ centered at ~4 and ~40 nm was observed with the D216A/D220A MRS2$_{58-333}$ protein (Fig 7C), similar to WT.

Together, these light scattering analyses demonstrate that Mg$^{2+}$-dependent disassembly of the MRS2 NTD requires the D216 and

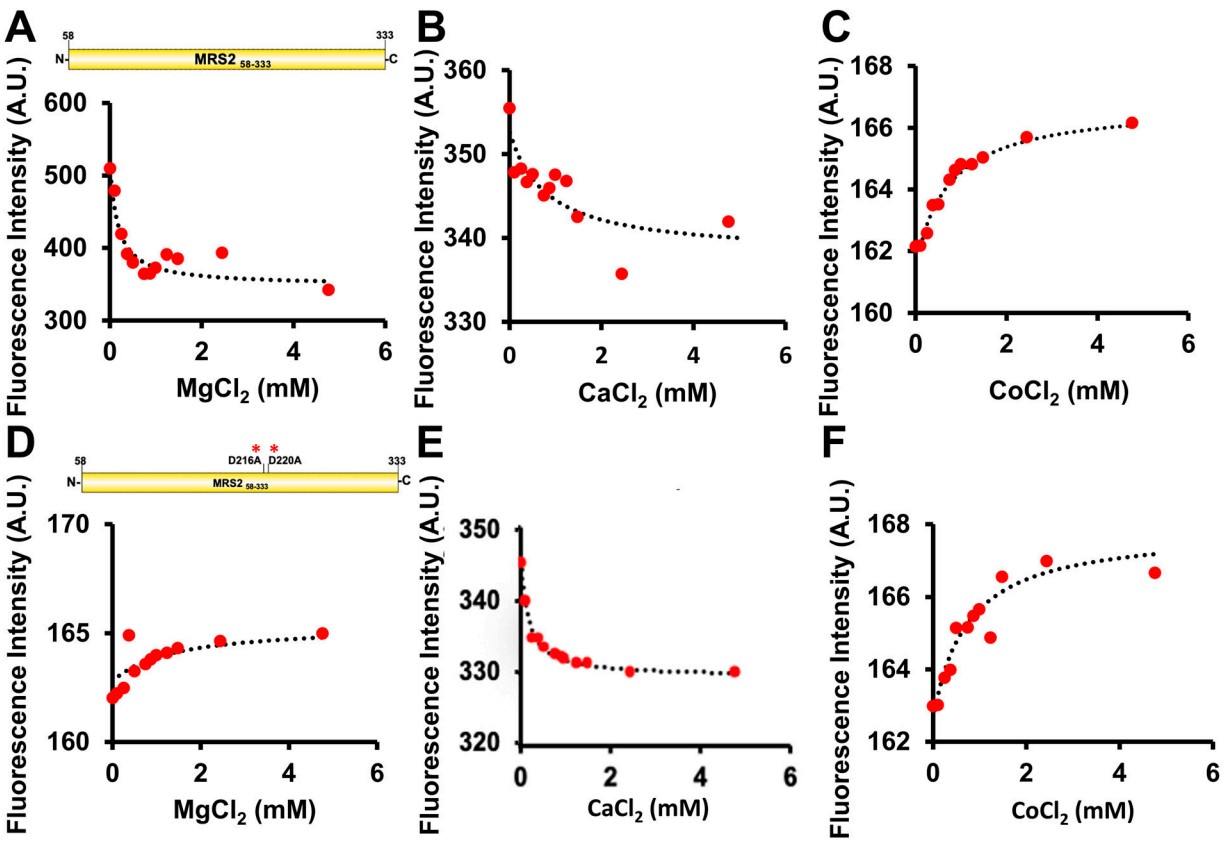

**Figure 5. Divalent cation binding affinity to MRS2$_{58-333}$ and MRS2$_{58-333}$ D216A/D220A (NTD).**
**(A)** Changes in intrinsic fluorescence emission intensity of MRS2$_{58-333}$ as a function of increasing MgCl$_2$ concentration. **(B)** Changes in intrinsic fluorescence emission intensity of MRS2$_{58-333}$ as a function of increasing CaCl$_2$ concentration. **(C)** Changes in intrinsic fluorescence emission intensity of MRS2$_{58-333}$ as a function of increasing CoCl$_2$ concentration. **(D)** Changes in intrinsic fluorescence emission intensity of MRS2$_{58-333}$ D216A/D220A as a function of increasing MgCl$_2$ concentration. **(E)** Changes in intrinsic fluorescence emission intensity of MRS2$_{58-333}$ D216A/D220A as a function of increasing CaCl$_2$ concentration. **(F)** Changes in intrinsic fluorescence emission intensity of MRS2$_{58-333}$ D216A/D220A as a function of increasing CoCl$_2$ concentration. In (A, B, C, D, E, F), data (red circles) show intrinsic fluorescence intensities at 330 nm as a function of increasing divalent cation concentration and are representative of n = 3 separate experiments (Table S3) performed from three protein preparations. The dashed lines through the data fit to a one-site binding model that accounts for protein concentration. All experiments were performed with 0.1 μM protein in 20 mM Tris, 150 mM NaCl, and 1 mM DTT, pH 8.0, at 22.5°C.

D220 residues, where double mutation to Ala abrogates quaternary structure sensitivity to the cation.

### D216A/D220A abrogates increased α-helicity and thermal stability in the MRS2 NTD caused by Mg$^{2+}$ binding

Having observed that Mg$^{2+}$ binding affects the quaternary and tertiary levels of MRS2 NTD structure, we next used far-UV circular dichroism (CD) spectroscopy to assess the secondary structure. At 37°C, MRS2$_{58-333}$ displayed well-defined mean residue ellipticity minima at ~208 and ~222 nm, indicating high levels of α-helicity (Fig 8A). Remarkably, addition of 5 mM MgCl$_2$ directly to the cuvette resulted in an increase in α-helicity, evidenced by more intense negative ellipticity at ~208 and 222 nm (Fig 8A and C). Similar results were observed for MRS2$_{58-333}$ at 20°C (Fig S4A).

To gain further evidence that D216 and D220 play a critical role in Mg$^{2+}$ binding to the NTD, we also acquired far-UV CD spectra using D216A/D220A MRS2$_{58-333}$. The far-UV CD spectrum of the double mutant showed a similar level of negative ellipticity as WT with two well-defined minima at ~208 and ~222 nm (Fig 8A and B), suggesting

that secondary structure folding was not perturbed by the D216A/D220A substitutions. Unlike WT, adding 5 mM MgCl$_2$ directly to the cuvette did not significantly alter the ellipticity for the double mutant (Fig 8D). Unchanging spectra after 5 mM MgCl$_2$ addition were also observed for the double mutant at 20°C (Fig S4B).

We next evaluated thermal stability by monitoring the change in far-UV CD ellipticity at 222 nm as a function of increasing temperature. The thermal melts of MRS2$_{58-333}$ acquired in the absence of Mg$^{2+}$ exhibited a mean Boltzmann sigmoidal–fitted midpoint of temperature denaturation ($T_m$) of 51 ± 0.62 °C (Fig 8E). Protein samples supplemented with 5 mM MgCl$_2$ were stabilized by ~7°C as the mean $T_m$ shifted to 58 ± 0.36°C (Fig 8E). Thermal melt experiments with the D216A/D220A MRS2$_{58-333}$ protein revealed similar mean $T_m$ values of 52 ± 0.70°C and 52 ± 0.64°C in the presence and absence of Mg$^{2+}$, respectively (Fig 8F).

Collectively, these data reveal that Mg$^{2+}$ binding stabilizes the MRS2 NTD, consistent with an observed increase in α-helicity. Furthermore, the structural and stability augmentation is dependent on D216 and D220 as mutation of these residues renders the NTD insensitive to Mg$^{2+}$, reinforcing the importance of these sites to coordinating Mg$^{2+}$.

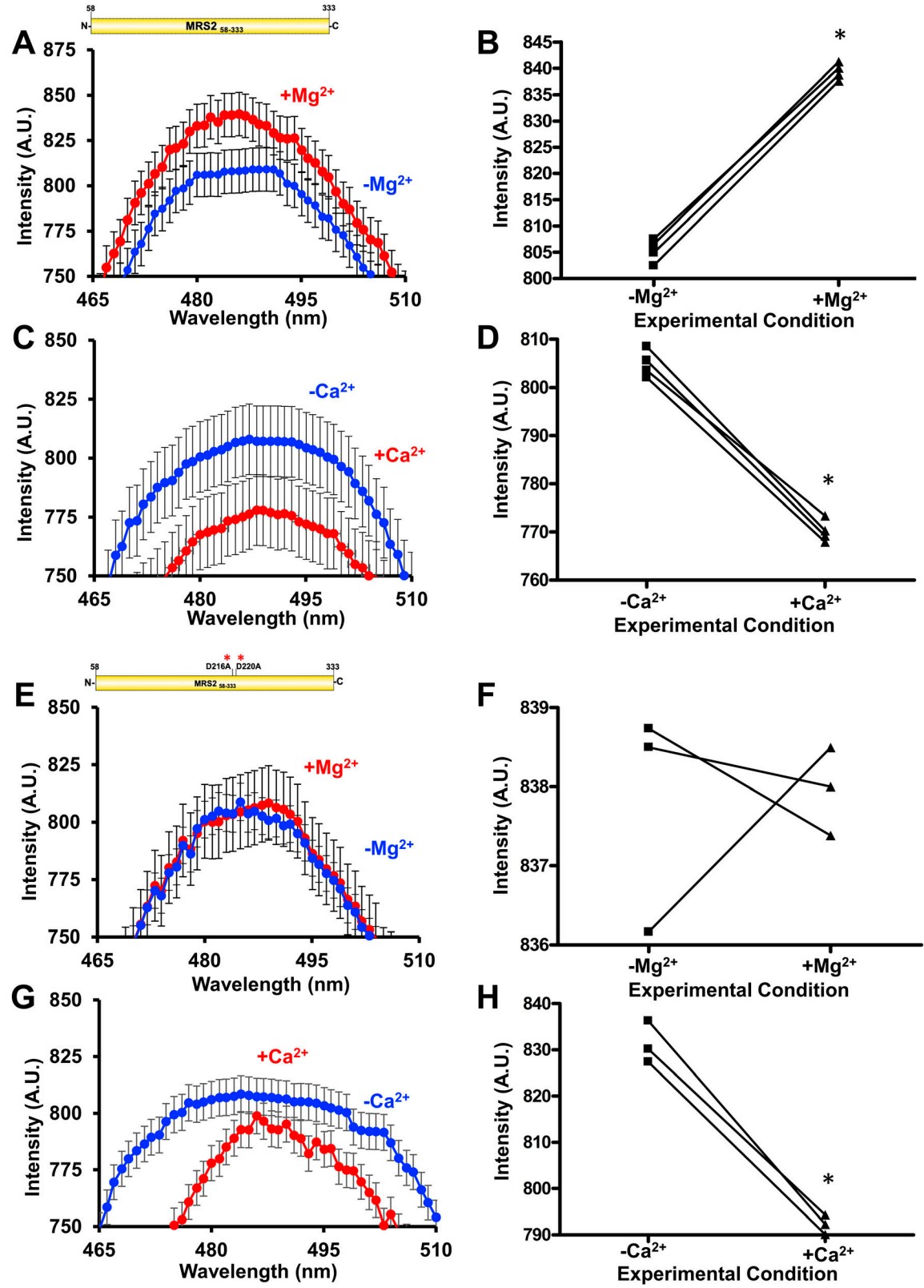

**Figure 6.   Solvent-exposed hydrophobicity of MRS2_{58–333} and MRS2_{58–333} D216A/D220A (NTD).**
**(A, B)** ANS fluorescence emission spectra of (A) MRS2$_{58–333}$ in the absence (blue) and presence (red) of 5 mM MgCl$_2$ and (B) paired statistical comparisons of the peak intensities. **(C, D)** ANS fluorescence emission spectra of (C) MRS2$_{58–333}$ in the absence (blue) and presence (red) of 5 mM CaCl$_2$ and (D) paired statistical comparisons of the peak intensities. **(E, F)** ANS fluorescence emission spectra of (E) MRS2$_{58–333}$ D216A/D220A in the absence (blue) and presence (red) of 5 mM MgCl$_2$ and (F) paired statistical

## Mg$^{2+}$ binding to the MRS2 NTD negatively regulates mitochondrial Mg$^{2+}$ uptake

To link our in vitro observations with MRS2 function, we monitored Mg$^{2+}$ dynamics using Mag-Green in HeLa cells overexpressing WT and D216A/D220A MRS2. HeLa cells were incubated with the membrane-permeant Mag-Green-AM to cytosolically load the cells with the Mg$^{2+}$ sensitive dye. After washing and bathing the cells with intracellular buffer (IB), the plasma membrane (PM) was permeabilized with 5 $\mu$M digitonin, and 3 mM MgCl$_2$ was added to the bath. Mitochondrial Mg$^{2+}$ uptake rates were inferred from the clearance of extramitochondrial Mg$^{2+}$, measured as the decrease in Mag-Green fluorescence, as previously done (Daw et al, 2020). After MgCl$_2$ addback, digitonin-permeabilized HeLa cells transfected with empty pCMV vector (control), pBSD-MRS2 (WT), and pBSD-MRS2 D216A/D220A (mutant), all showed increases in Mag-Green fluorescence followed by a decay associated with Mg$^{2+}$ clearance (Fig 9A). Fitting the data to single exponential decays indicated greater extramitochondrial Mg$^{2+}$ clearance rates for WT MRS2-expressing cells compared with control cells and mutant MRS2-expressing cells compared with control and WT MRS2-expressing cells (Fig 9B and C). Addition of 10 mM or 30 mM NaCl to similarly permeabilized cells caused no change in the Mag-Green signal, suggesting minimal influence of osmolarity on our Mag-Green measurements (Fig S5A and B).

Collectively, these data suggest that MRS2 overexpression enhances extramitochondrial Mg$^{2+}$ clearance, and Mg$^{2+}$ interactions with the MRS2 NTD act as a negative feedback switch to temper Mg$^{2+}$ uptake into the mitochondria.

## Gain of function D216K/D220K mutant relieves negative feedback on MRS2 activity

To probe whether mutation of the Mg$^{2+}$-binding site causes a bona fide gain of function, we reconstituted human WT and D216K/D220K MRS2 in WT and Mrs2 knockout (Mrs2$^{-/-}$) hepatocytes. Primary murine hepatocytes were transfected with empty vector, human MRS2-mRFP, or MRS2 D216K/D220K-mRFP plasmids. 24 h post-transfection, a genetically encoded, mitochondrially targeted Mag-FRET biosensor (i.e., mito-Mag-FRET) was transduced into the cells to directly measure mitochondrial Mg$^{2+}$ uptake. This mito-Mag-FRET sensor was previously shown to localize to hepatocyte mitochondria, reporting reciprocal lactate-induced Mg$^{2+}$ responses compared with an ER-targeted/retained Mag-FRET sensor (Daw et al, 2020). Murine Mrs2-mRFP under the control of a CMV promoter, similar to the construct used in the present study, was also shown to properly co-localize with dihydrorhodamine-123 in hepatocyte mitochondria (Daw et al, 2020). Here, confocal images of the transfected/transduced WT hepatocytes show strong co-expression and co-localization of MRS2 and MRS2 D216K/D220K with the cerulean and citrine fluorescence of the mito-Mag-FRET

biosensor, indicating mitochondrial localization of the human WT and mutant MRS2 in murine cells (Figs 10A and S6A). The pixel intensity profiles of the mito-Mag-FRET citrine and human MRS2-mRFP (WT and mutant) signals exhibit coincident peak maxima, consistent with this co-localization (Fig S7A and B). Human WT and mutant MRS2 also strongly co-localized with the mito-Mag-FRET fluorophores in the Mrs2$^{-/-}$ hepatocytes (Figs 10B and S6B).

As expected, a 10-mM MgCl$_2$ bolus increased the mito-Mag-FRET signal in WT cells (Fig 10C). Although WT hepatocytes expressing human MRS2 showed a similar mito-Mag-FRET response to the MgCl$_2$ bolus, cells transfected with human MRS2 D216K/D220K exhibited highly potentiated mitochondrial Mg$^{2+}$ uptake compared with controls (Fig 10D). Human MRS2 was fully capable of functionally reconstituting the Mg$^{2+}$ channel in Mrs2$^{-/-}$ hepatocyte mitochondria. Remarkably, the MRS2 D216K/D220K formed channels that greatly enhanced mitochondrial Mg$^{2+}$ uptake compared with WT human MRS2 (Fig 10E). Note that the mRFP fluorescence intensities of human WT MRS2 and human MRS2 D216K/D220K in WT and Mrs2$^{-/-}$ hepatocytes were similar, suggesting comparable expression levels across all groups (Fig S6C). Given the striking potentiation of Mg$^{2+}$ uptake in hepatocytes co-expressing the D216K/D220K mutant but not WT human MRS2 with endogenous Mrs2, our data suggest that the Mg$^{2+}$ binding–deficient MRS2 mutant dominantly mediates a gain of mitochondrial Mg$^{2+}$ uptake function.

# Discussion

Human MRS2 belongs to the heterogeneous CorA/Mrs2/Alr1 superfamily of Mg$^{2+}$ transporters, where CorA, Alr1, and Mrs2/MRS2 comprise the principal Mg$^{2+}$ uptake systems in bacteria, yeast PM, and mitochondria, respectively. Bacterial CorA has been the most extensively studied family member, yielding mechanistic and functional insights on these channels (Franken et al, 2022; Jin et al, 2022). Nevertheless, given the low sequence similarity between human MRS2 and these homologues, there remains a major knowledge gap concerning the precise structural, functional, and regulatory mechanisms of human MRS2. Here, we isolated and biophysically characterized the largest domain of human MRS2, corresponding to the matrix-oriented NTD. We found that MRS2 NTD forms a homodimer under dilute conditions, which may be a building block to higher order oligomers. Remarkably, Mg$^{2+}$ and Ca$^{2+}$ disassembled both higher order MRS2 NTD oligomers and homodimers but not full-length MRS2 assemblies. In contrast, Co$^{2+}$ disassembled full-length MRS2 oligomers but not MRS2 NTD. We estimated the K$_d$ of Mg$^{2+}$ binding to be ~0.14 mM, and a D216A/D220A MRS2 NTD double mutant disrupted this Mg$^{2+}$ binding but had no effect on Ca$^{2+}$ binding, indicating disparate binding sites for these two divalent cations. Remarkably, this D216A/D220A double mutant abrogated the enhanced solvent exposed hydrophobicity,

comparisons of the peak intensities. **(G, H)** ANS fluorescence emission spectra of (G) MRS2$_{58-333}$ D216A/D220A in the absence (blue) and presence (red) of 5 mM CaCl$_2$ and (H) paired statistical comparisons of the peak intensities. In (A, C, E, G), data are means ± SEM of n = 3 separate samples from three protein preparations. In (B, D, F, H), comparisons are paired $t$ test analyses, where *$P$ < 0.05. ANS binding experiments were performed using 30 $\mu$M protein and 50 $\mu$M ANS in 20 mM Tris, 150 mM NaCl, and 1 mM DTT, pH 8.0, at 15°C.

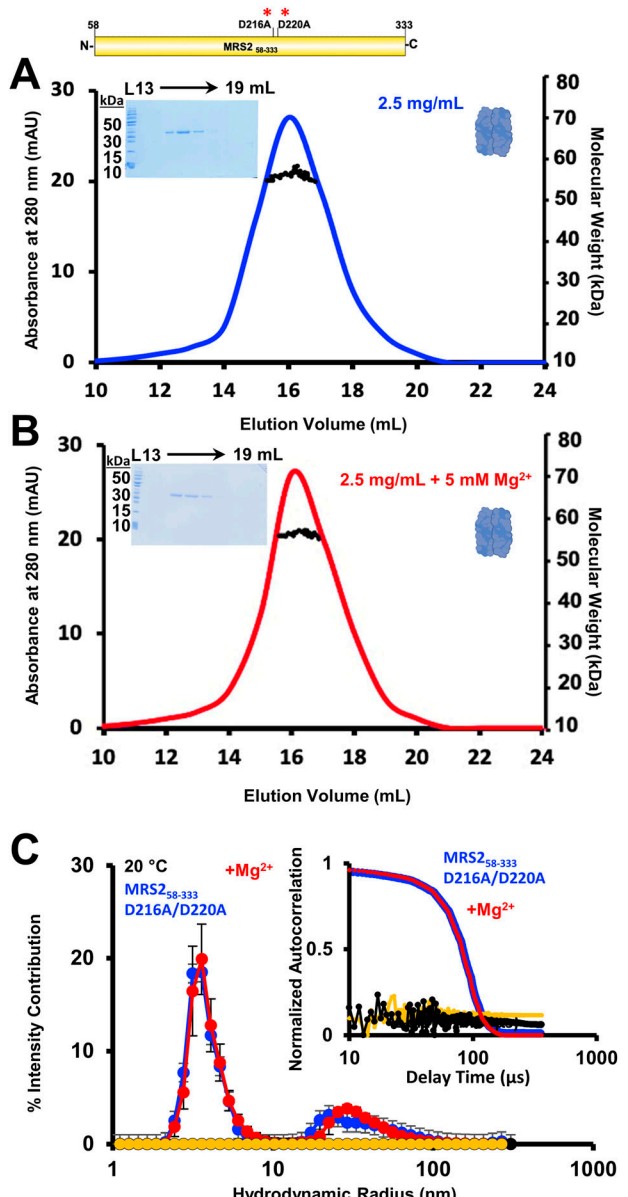

**Figure 7. Quaternary structure and higher order oligomerization of MRS2₅₈₋₃₃₃ D216A/D220A (NTD).**

**(A, B)** SEC-MALS analysis of MRS2$_{58-333}$ D216A/D220A injected at 2.5 mg/ml in the (A) absence or (B) presence of 5 mM MgCl$_2$. **(C)** DLS analysis of MRS2$_{58-333}$ D216A/D220A at 1.25 mg/ml. The distributions of R$_h$ from the regularization deconvolution of the autocorrelation functions are shown in the presence and absence of 5 mM MgCl$_2$ at 20°C. In (A, B), MALS-determined molecular weights are shown through the elution peaks (black circles), left insets show Coomassie blue–stained SDS–PAGE gels of the elution fractions from the 2.5 mg/ml injections and right insets depict the dimerization state of the protein. Elution volumes are indicated at top and ladder "L" molecular weights at left of the gels. Data are representative of n = 3 separate injections from three protein preparations (Table S1) and were acquired using an S200 10/300 Gl column in 20 mM Tris, 150 mM NaCl, and 1 mM DTT, pH 8.0, 10°C. In (C), inset show the Mg$^{2+}$-induced shift in the autocorrelation functions, Mg$^{2+}$-free protein sample data are coloured blue, Mg$^{2+}$-supplemented protein sample data are red, Mg$^{2+}$-free buffer control data are black, and Mg$^{2+}$-supplemented buffer control data are yellow. Inset data are representative, while deconvoluted R$_h$ profiles are means ± SEM of n = 3 separate samples from three protein preparations. Data were acquired in 20 mM Tris, 150 mM NaCl, and 1 mM DTT, pH 8.0, 20°C.

α-helicity, and thermal stability mediated by Mg$^{2+}$ binding. Furthermore, MRS2 NTD oligomers and homodimers harboring this double mutation remained intact in the presence of Mg$^{2+}$. Finally, we showed that reconstitution of D216A/D220A or D216K/D220K MRS2 mutants in mammalian cells greatly increased mitochondrial Mg$^{2+}$ uptake compared with WT MRS2-expressing cells.

Several CorA crystal and cryoelectron microscopy structures have been elucidated in the presence of divalent cations, revealing a pentameric assembly (Eshaghi et al, 2006; Payandeh & Pai, 2006; Guskov et al, 2012; Pfoh et al, 2012; Nordin et al, 2013; Cleverley et al, 2015; Matthies et al, 2016; Johansen et al, 2022). The first TM, which lines the channel pore, and second TM orient the intervening GMN motif for ion binding and selectivity at the pore entrance (Pfoh et al, 2012). Upstream of TM1, a large intracellular domain of CorA, analogous to the matrix-oriented human MRS2 NTD, fans out into the cytoplasm and is composed of eight α-helices and a six-stranded β-sheet (T. maritima; 4EED.pdb) (Fig 11A). For T. maritima CorA, two Mg$^{2+}$-binding sites (M1 and M2) have been identified per intracellular domain (Pfoh et al, 2012). M1 is made up of D89 and D253, whereas M2 is comprised of D175 and D179. Whereas earlier studies indicated that a symmetrizing of the pentameric intracellular domain assembly upon Mg$^{2+}$ binding to the NTD closes the channel (Pfoh et al, 2012; Matthies et al, 2016), more recent work indicates both symmetric and asymmetric assemblies are formed in the presence and absence of Mg$^{2+}$, and channel conductance is dependent on lowered symmetric state population and coupled with a reduced energy barrier to an ensemble of open states in low Mg$^{2+}$ (Kowatz & Maguire, 2019; Johansen et al, 2022). Nevertheless, it is evident that Mg$^{2+}$ binding increases the rigidity/decreases the dynamics of the CorA intracellular domain (Chakrabarti et al, 2010; Pfoh et al, 2012; Rangl et al, 2019; Johansen et al, 2022).

The M1 Mg$^{2+}$-binding site of T. maritima CorA does not appear conserved in human MRS2 based on multiple sequence alignments (Figs S1 and S2) or 3D superposition of the CorA crystal (Pfoh et al, 2012) and human AlphaFold2 (Jumper et al, 2021) MRS2-predicted structures. Furthermore, the AlphaFold2 model of human MRS2 orients the D216 and D220 Mg$^{2+}$-binding residues, which we experimentally validated, six helical turns closer to the membrane domains compared with the CorA M2 DALVD residues (Fig 11A and B). Interestingly, a superposition of T. maritima CorA and S. cerevisiae Mrs2 (Khan et al, 2013) crystal structures structurally aligns the bacterial CorA DALVD with an INVMS sequence in S. cerevisiae Mrs2 (Fig S8A), suggesting the bacterial M2 Mg$^{2+}$-binding site is not conserved in yeast. However, a superposition of the human AlphaFold2 model with the S. cerevisiae crystal structure suggests a structural conservation between human D216 and D220 and yeast D203 and E207 (of DLENE) (Fig S8B), not apparent from the multiple sequence alignments (Figs S1 and S2).

Although the stoichiometry of the human MRS2 channel remains unknown, we generated a homopentamer in homology to CorA, using AlphaFold-Multimer (Evans et al, 2022 Preprint) to model how Mg$^{2+}$ binding to D216 and D220 may cause matrix domain disassembly (Fig 11B). PDBsum analysis (Laskowski et al, 2018) of the multimer model indicates that D216 and D220 do not participate in interprotomer H-bonding, salt-bridges, or other nonbonded contacts. However, P221 H-bonds and forms other nonbonded contacts

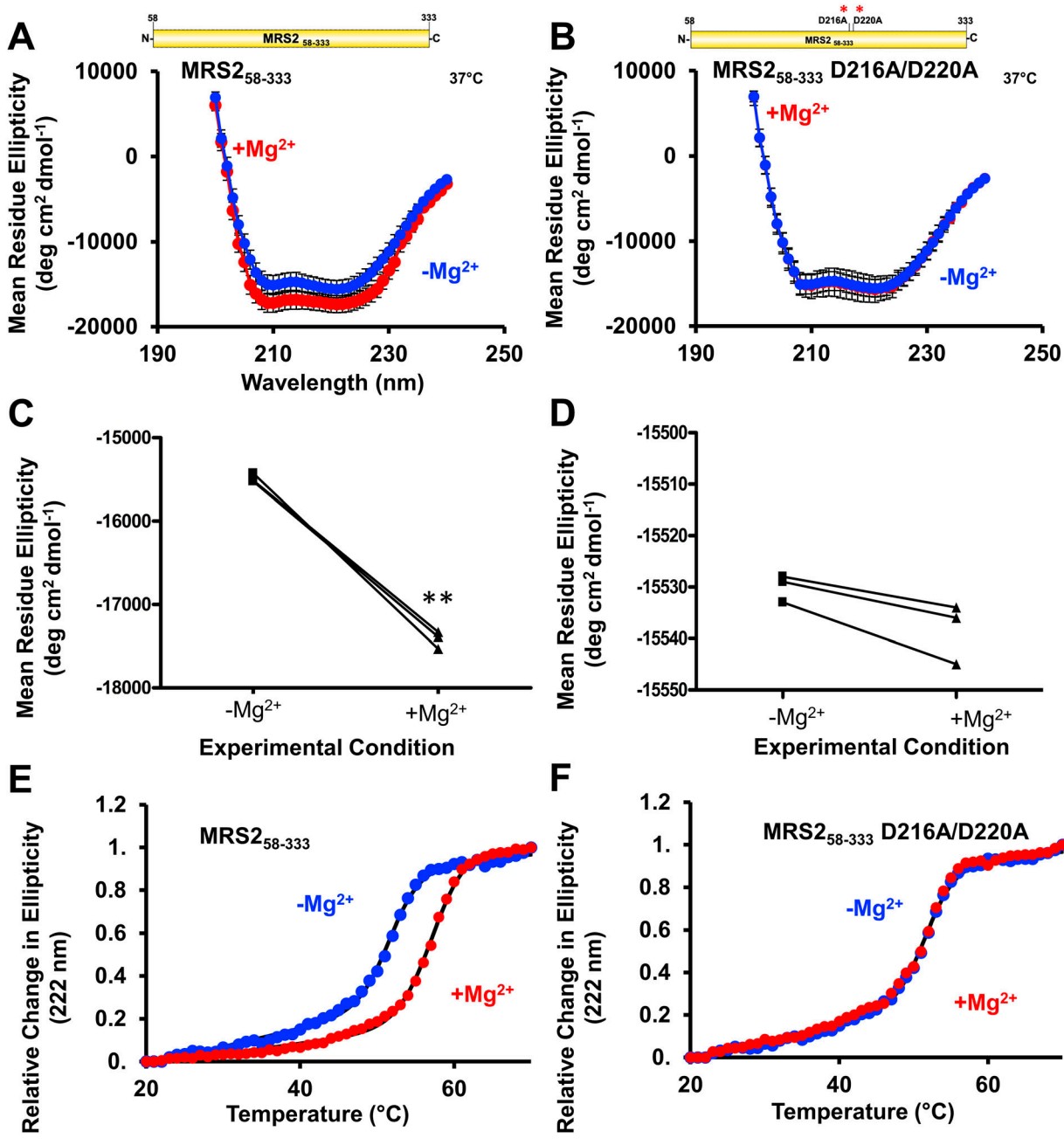

**Figure 8. Secondary structure and thermal stability of MRS2₅₈₋₃₃₃ and MRS2₅₈₋₃₃₃ D216A/D220A (NTD).**
**(A, B)** Far-UV CD spectra of (A) MRS2$_{58–333}$ and (B) MRS2$_{58–333}$ D216A/D220A in the absence (blue) and presence (red) of 5 mM MgCl$_2$. **(C, D)** Statistical comparisons of Mg$^{2+}$-induced changes in mean residue ellipticity at 222 nm for (C) MRS2$_{58–333}$ and (D) MRS2$_{58–333}$ D216A/D220A. **(E, F)** Changes in mean residue ellipticity (222 nm) as a function of increasing temperature (i.e., thermal stability) in the absence (blue) and presence (red) of 5 mM MgCl$_2$ for € MRS2$_{58–333}$ and (F) MRS2$_{58–333}$ D216A/D220A. In (A, B), data are means ± SEM of n = 3 experiments with samples from three protein preparations. In (A, B, C, D), comparisons are paired $t$ test analyses of data from (A, B), where $**P < 0.01$. In (E, F), data (circles) are representative of n = 3 separate experiments with samples from three protein preparations, and solid black lines are Boltzmann sigmoidal fits through the data to extract apparent midpoints of temperature denaturation (T$_m$). All far-UV CD experiments were acquired with 0.5 mg/ml protein in 20 mM Tris, 150 mM NaCl, and 1 mM DTT, pH 8.0, with spectra acquired at 37°C.

with R228 of an adjacent subunit (Fig 11C). Furthermore, the loop following D220 exits into a 32-residue helix that contains many additional interprotomer contacts (Fig 11C). We showed that Mg$^{2+}$ binding to D216 and D220 increases α-helicity, which could rearrange the adjacent loop that contains P221 and the position of the immediate downstream helix, leading to subunit dissociation. The human MRS2 homopentamer model also reveals clusters of negatively charged residues across interfaces (i.e., E243, D247, D305, and E312), which could mediate additional divalent cation–binding sites (Fig 11C). Ultimately, experimentally determined high-resolution

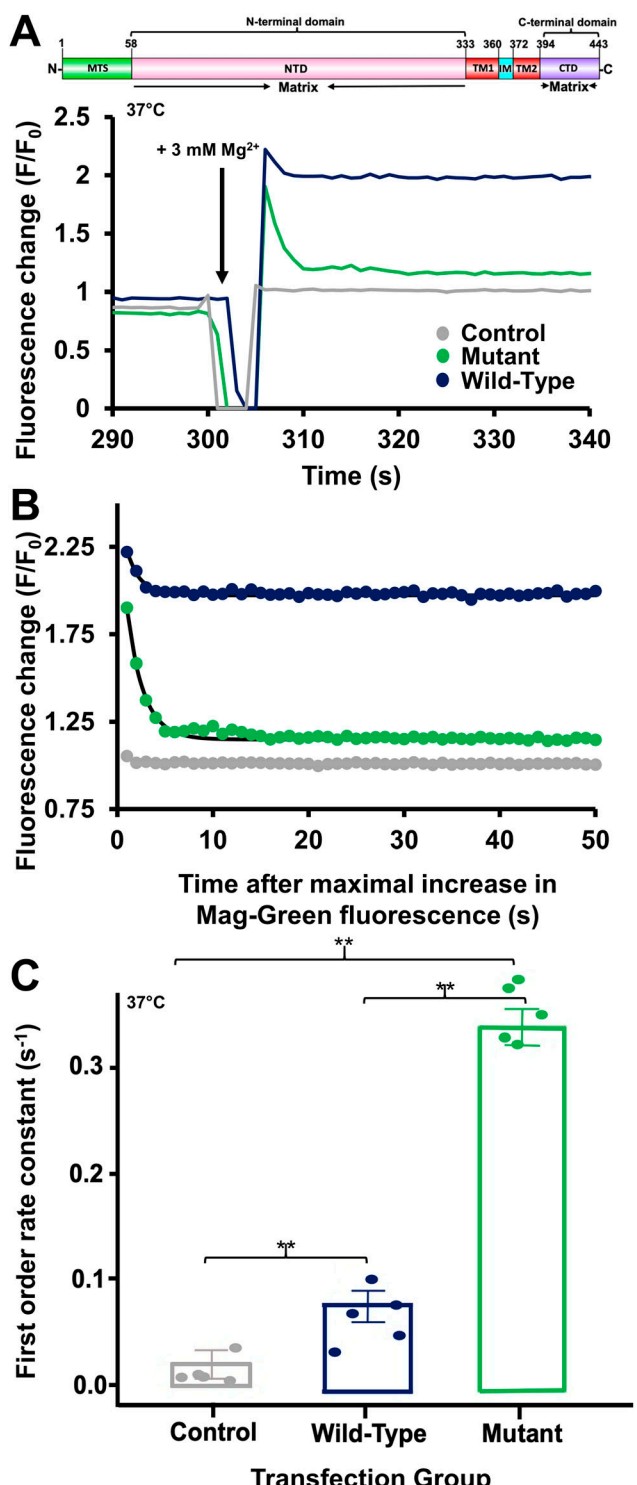

**Figure 9. Mitochondrial Mg²⁺ uptake rates in HeLa cells overexpressing WT or D216A/D220A human MRS2.**
**(A)** Representative Mag-Green fluorescence traces reporting relative extramitochondrial Mg²⁺ before and after 3 mM MgCl₂ addback to the extracellular bath at 300 s (black arrows) for control-, WT MRS2–, and MRS2 D216A/D220A–transfected cells. **(A, B)** Representative single exponential fits to the Mag-Green fluorescence decays after the 3 mM MgCl₂ addback shown in (A), reporting Mg²⁺ clearance and taken as a measure of mitochondrial Mg²⁺ uptake. **(C)** One-way ANOVA followed by Tukey's post hoc comparison of mitochondrial Mg²⁺

structures of human MRS2 are needed to reveal channel stoichiometry, the basis for assembly and mechanisms for Mg²⁺-induced disassembly.

Mg²⁺ increased α-helicity and stability of the human MRS2 NTD, consistent with past NMR data, showing decreased backbone dynamics of CorA in the presence of high Mg²⁺ (Johansen et al, 2022). The far-UV CD spectra reported here resemble previous data from our laboratory, where we found no effect by MgCl₂, likely due to variability in protein concentration measurements (Daw et al, 2020). Here, we applied MgCl₂ addback to the same sample to expose the secondary structure change. Several lines of evidence suggest distinct Ca²⁺- and Mg²⁺-binding sites on the MRS2 NTD. First, the change in intrinsic fluorescence caused by the two cations was different; second, whereas Mg²⁺ increased, Ca²⁺ decreased solvent accessible hydrophobicity; third, D216A/D220A double mutant increased the Mg²⁺ $K_d$ ~sevenfold, whereas having no effect on the Ca²⁺ $K_d$; finally, the Mg²⁺-dependent solvent accessible hydrophobicity change was abrogated, whereas the Ca²⁺ response was maintained by the D216A/D220A double mutant. Given the MM has a free Mg²⁺ concentration of ~0.5–1.5 mM (Jung et al, 1990; Rutter et al, 1990), the Mg²⁺ $K_d$ of ~0.14 mM reported here would suggest the MRS2 structure, stability, and oligomerization would be sensitive to physiologically relevant fluctuations in Mg²⁺ levels within the matrix.

A broad range of free MM Ca²⁺ concentrations in mammalian cells have been reported, dependent on cell type, stimulus, and indicator; moreover, most estimates are < 100 μM (reviewed in Fernandez-Sanz et al [2019]), much lower than our MRS2 NTD Ca²⁺ $K_d$ estimate of ~1 mM. Hence, the MRS2 NTD would have to be positioned close to a Ca²⁺ channel pore to be affected, where local Ca²⁺ concentrations may approach the ~mM range (Chad & Eckert, 1984; Bauer, 2001; Tadross et al, 2013). Directly assessing how Ca²⁺ binding affects MRS2 activity in cellulo is problematic due to the weak Ca²⁺ $K_d$ of 1 mM. For example, 100 μM matrix Ca²⁺ would occupy < 10% of the MRS2 Ca²⁺-binding sites and perturb mitochondrial membrane potential. Nevertheless, indirectly, mitochondrial Ca²⁺ uniporter KO studies show unaltered lactate-stimulated mitochondrial Mg²⁺ uptake (Daw et al, 2020), suggesting Ca²⁺ may not play a crucial role in MRS2 regulation.

Interestingly, although Co²⁺ dissociated larger full-length MRS2 assemblies, we observed no effect on MRS2 NTD by DLS. In contrast, Mg²⁺ and Ca²⁺ did not alter the assembly of full-length MRS2 but dissociated the MRS2 NTD. We posit that MRS2₅₈₋₃₃₃ (NTD) within MRS2 full-length undergoes disassembly in the presence of Mg²⁺ and Ca²⁺, whereas the C-terminal domain and/or TM regions remain interacting. Such a change would be undetectable by DLS, as the complex size would be unaffected. Furthermore, we believe Co²⁺-mediated disassembly occurs via binding to a region outside the NTD. Because estimates for the mitochondrial concentration of Co²⁺

uptake rates for control-, WT MRS2–, and MRS2 D216A/D220A–transfected cells, where *$P$ < 0.05, **$P$ < 0.01, and ***$P$ < 0.001. In (A, B, C), data were acquired at 37°C and were normalized as F/F₀, where F is the Mag-Green fluorescence at any time point and F₀ is the mean 30 s baseline fluorescence before the addition of EDTA/digitonin, and control, WT, and D216A/D220A data are coloured grey, black, and green, respectively.

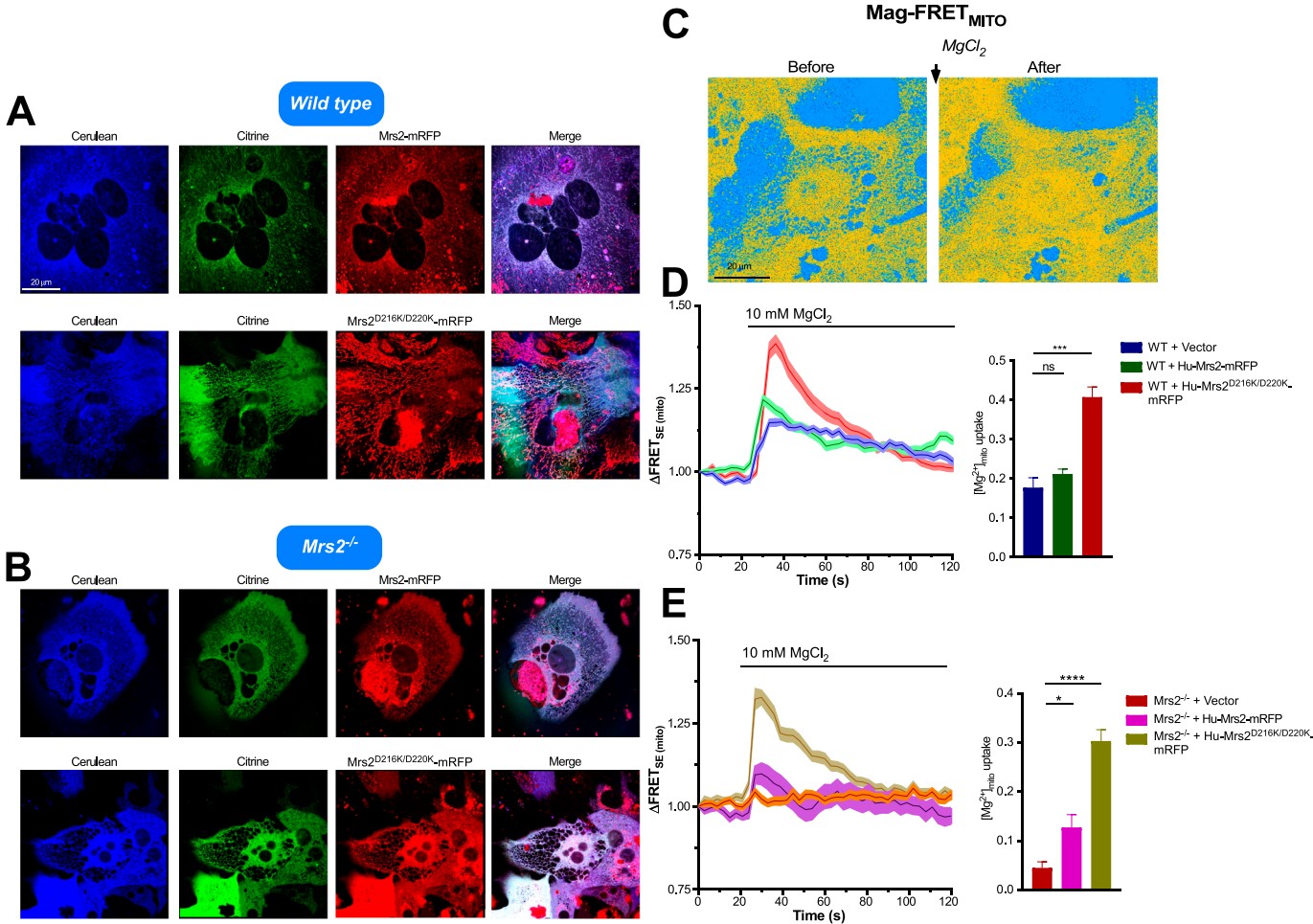

**Figure 10. D216K/D220K double mutant enhances human MRS2 channel activity.**
**(A, B)** Representative confocal images showing expression and mitochondrial localization of human WT MRS2-mRFP and MRS2 D216K/D220K-mRFP in WT (A) or Mrs2$^{-/-}$ (B) primary murine hepatocytes transduced with the adenoviral mito-Mag-FRET sensor. **(C)** Representative FRET images before and after 10 mM MgCl$_2$ addition to primary hepatocytes expressing mito-Mag-FRET. **(D)** Mean traces (left panel) showing relative mito-Mag-FRET ratio changes upon addition of 10 mM MgCl$_2$ and comparisons of peak mitochondrial Mg$^{2+}$ uptake responses in WT hepatocytes (right panel). **(E)** Mean traces (left panel) showing relative mito-Mag-FRET ratio changes upon addition of 10 mM MgCl$_2$ and comparisons of peak mitochondrial Mg$^{2+}$ uptake responses in Mrs2$^{-/-}$ hepatocytes (right panel). In (A, B, C, D, E), all measurements were performed at 37°C and quantified values are normalized as FRET/FRET$_0$, where FRET and FRET$_0$ are the signals at any time and initial timepoints, respectively. Unpaired $t$ test was performed for comparison of peak responses, where *$P < 0.05$, ***$P < 0.001$, ****$P < 0.0001$; ns, not significant. Data are means ± SEM of n = 3 for WT + vector, n = 5 for WT + Hu-MRS2-mRFP, n = 3 for WT + Hu-MRS2$^{\Delta216K/D220K}$-mRFP, n = 3 for Mrs2$^{-/-}$ + vector, n = 4 for Mrs2$^{-/-}$ + Hu-MRS2-mRFP, and n = 3 for Mrs2$^{-/-}$ + Hu-MRS2$^{\Delta216K/D220K}$-mRFP, where n is the number of separate transfections.

range from ~50 to 90 nM (Tapiero et al, 2003; Czarnek et al, 2015), the precise physiological significance of Co$^{2+}$ interactions with any human MRS2 domain remains unclear.

Using permeabilized and intact cells, our data show that Mg$^{2+}$ binding to the MRS2 NTD negatively regulates the channel. Permeabilized cells overexpressing the D216A/D220A double-mutant MRS2 cleared extramitochondrial Mg$^{2+}$ at increased rates compared with WT MRS2-expressing cells. Furthermore, human WT and D216K/D220K MRS2 were fully capable of reconstituting functional MRS2 channels in intact primary murine Mrs2$^{-/-}$ hepatocytes, with the double mutant causing highly potentiated Mg$^{2+}$ uptake in Mrs2$^{-/-}$ and WT mitochondria, indicative of gain of function activity. We do not believe that osmolarity changes because of MgCl$_2$ addition influenced these trends since construct-specific responses were observed and 10 mM or 30 mM NaCl had no effect on the Mag-Green

responses (Fig S5). Interestingly, a study using CorA harboring mutations aimed at disrupting Mg$^{2+}$ binding to M1 showed WT-like $^{63}$Ni$^{2+}$ transport (Kowatz & Maguire, 2019). Here, we focused on an M2-like cluster of residues because M1 does not appear to be conserved in human MRS2, discovering a robust, dominantly increased mitochondrial Mg$^{2+}$ uptake upon disruption of Mg$^{2+}$ binding to the NTD.

In conclusion, our work reveals the large NTD functions as a negative feedback regulator of human MRS2 channel function. We propose Mg$^{2+}$ binding to the MRS2 NTD, contributed by D216 and D220, disrupts NTD:NTD interactions without disassembly of the channel (Fig 11D). Mg$^{2+}$ binding to the MRS2 NTD increases $\alpha$-helicity, stability, and solvent exposed hydrophobicity but dissociates NTD: NTD complexes, which we believe underlie key structural changes that propagate to the pore and/or crucial gating residues to inhibit

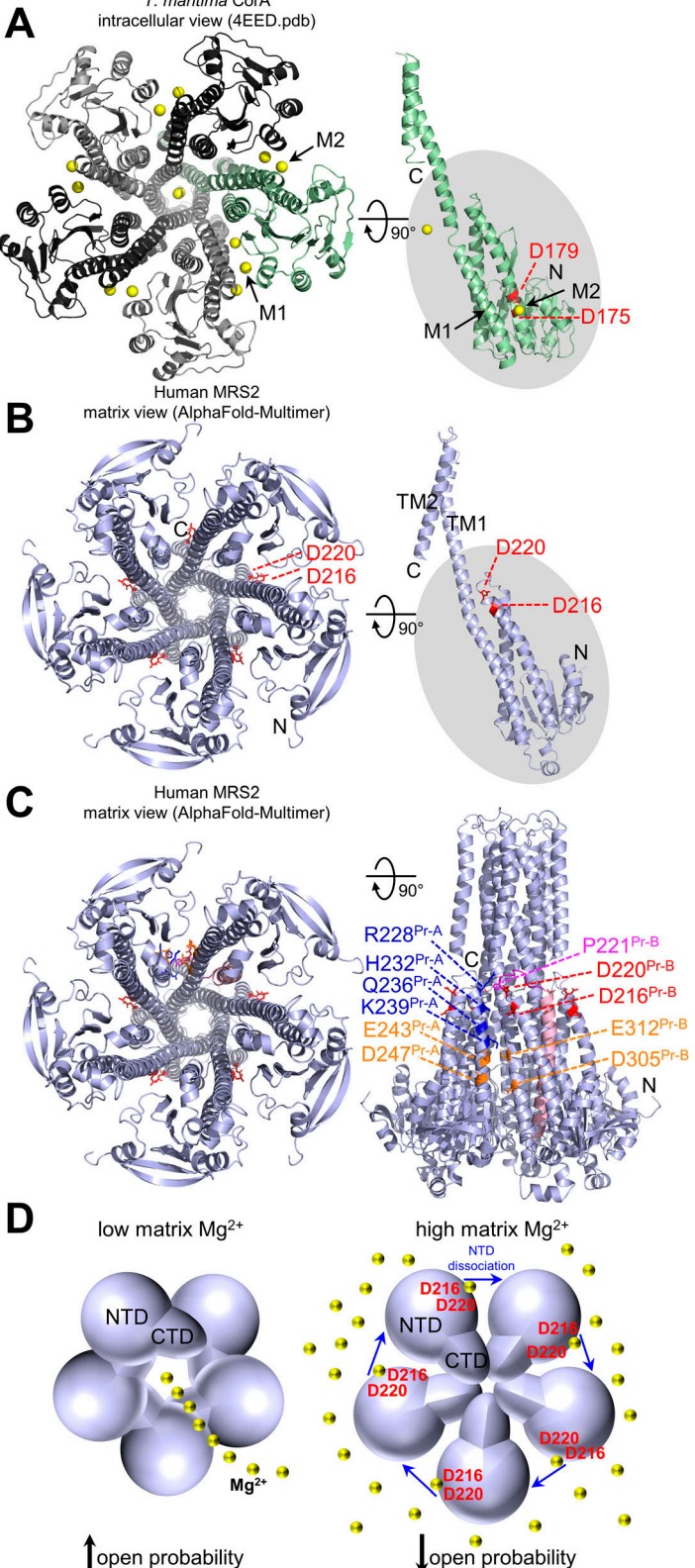

**Figure 11. Model of Mg²⁺-induced negative feedback in MRS2 function.**
**(A)** At left, intracellular view of the experimentally determined *T. maritima* CorA pentamer structure. A backbone cartoon representation of each protomer is shown with M1 and M2 Mg²⁺-binding sites indicated (black arrows). At right, view of a single protomer (green), rotated 90° relative to the pentamer view. The D175 and D179 of the bacterial DALVD are coloured red. **(B)** At left, matrix view of the AlphaFold-Multimer model of a human MRS2 homopentamer. At right, view of a single protomer (light blue) rotated 90° relative to the pentamer view. The D216 and D220 positions of the human DALVD are coloured red. **(C)** At left, the matrix view AlphaFold-Multimer model of a human MRS2 homopentamer. At right, view of the 90° rotated homopentamer, highlighting the position of the D216 and D220 Mg²⁺ binding residues (red), the adjacent loop (magenta), interprotomer P221 (magenta):R228 (blue) contacts, connected helix-mediating additional interprotomer contacts (light pink), additional potential metal-binding sites across protomers involving E243, D247, E312, and D305 (orange) and the location of H232, Q236, and K239 (blue) on an adjacent protomer near D216 and D220. Pr-A, protomer-A; Pr-B, protomer-B. **(D)** Model of matrix Mg²⁺-induced inhibition of human MRS2 open probability. At low matrix Mg²⁺ (left), the human MRS2 channel has a high open probability; at high matrix Mg²⁺ (right), Mg²⁺ binding to a site mediated by D216 and D220 dissociates NTD complexes, leading to a low open probability. Despite weakened NTD interactions at high Mg²⁺, the channel remains assembled via TM and/or CTD interactions. A pentameric human MRS2 channel is depicted in homology to CorA, but the functional stoichiometry of the human MRS2 channel remains unknown. In (A, B, C), the same relative view is highlighted by the grey oval, Mg²⁺ ions are represented by yellow spheres and the amino and carboxyl termini are labeled N and C, respectively. In (B, C), the DALVD and adjacent loop sequence show AlphaFold2 pLDDT confidence scores of < 90 and < 70, indicating the predicted orientations of the side chains and backbone, respectively, are not reliable. An experimentally determined human MRS2 structure is needed to firmly establish the precise stoichiometry and the positions of the D216 and D220 side chains as well as nearby residues with respect to the assembled channel. Source data are available for this figure.

the channel (Fig 11D). These data distinguish human MRS2 from bacterial CorA observations, where $Mg^{2+}$ binding to the analogous intracellular domain shields electrostatically repulsive interfaces, promoting and bridging a symmetric interaction between intra-cellular domains (Pfoh et al, 2012; Matthies et al, 2016).

Because D216 and D220 are predicted to be near H232, Q236, and K239 of an adjacent promoter (Fig 11C), the D216A/D220A mutation could potentially perturb inter–promoter interactions involving these residues. In this scenario, destabilization of inter-domain interactions could lead to increased human MRS2 channel open probability, similar to the $Mg^{2+}$ dissociation-dependent mechanism recently articulated in detail for *T. maritima* CorA by the A. Guskov group (Nemchinova et al, 2021). However, this scenario appears to be inconsistent with the $Mg^{2+}$-binding–induced monomerization we observed for the human MRS2 matrix domain and our observations that D216A/D220A does not alter the dimer stoichiometry of the human MRS2 matrix domain or the higher order full-length human MRS2 assembly in CHAPS micelles.

# Materials and Methods

### MRS2 expression and purification

The human MRS2 NTD was identified as residues 58–333 using bioinformatic identification of the mitochondrial targeting sequence (Fukasawa et al, 2015; Almagro Armenteros et al, 2019; Buchan & Jones, 2019), and TM1 and TM2 (Krogh et al, 2001), after the comparison with these predictions with the annotations in UniProt (Accession Q9HD23). Human $MRS2_{58-333}$ was subcloned out of the BDS vector into pET-28a (Novagen) using PCR and NdeI and XhoI restriction sites. Overnight protein expression at 37°C from the pET-28a-$MRS2_{58-333}$ vector was done using BL21 (DE3) *Escherichia coli* cells cultured in Luria broth, induced with 0.4 mM IPTG. Protein was purified under native conditions using HisPur (Thermo-Fisher Scientific) nickel–nitrilotriacetic acid beads as per the manufacturer guidelines. The wash and elution buffers contained 20 mM Tris (pH 8.0), 150 mM NaCl, 1 mM DTT, 20 mM Tris (pH 8.0), 150 mM NaCl, 1 mM DTT, and 300 mM imidazole. After dialysis in 20 mM Tris (pH 8.0), 150 mM NaCl, and 1 mM DTT buffer using a 3,500 D MWCO membrane (Thermo Fisher Scientific), the N-terminal hexa-histidine tag was cleaved with ~2 U of bovine thrombin (Sigma-Aldrich) per 1 mg of protein. A final SEC step through an S200 10/300 Gl column (Cytiva), achieved >95% protein purity as assessed by SDS–PAGE and Coomassie blue staining.

D216A and D220A mutant were introduced into $MRS2_{58-333}$ by PCR-mediated site-directed mutagenesis and expression and purification for this construct were performed as described for WT-$MRS2_{58-333}$. The complementary mutagenic primers were 5'-CCTTGAGACCTTGGCTGCTTTGGTGGCCCCCAAACATTCTTC-3' and 3'-GAAGAATGTTTGGGGGCCACCAAAGCAGCCAAGGTCTCAAGG-5'.

Full-length human MRS2 taken as residues 58–443 ($MRS2_{58-443}$) was cloned and expressed using the same approach described for $MRS2_{58-333}$. Purification was performed as described for the NTD, except with the addition of 10 mM CHAPS to both the elution and SEC buffers.

### SEC with in-line multi angle light scattering

SEC-MALS was performed using a Superdex 200 Increase 10/300 Gl column (Cytiva) connected to an AKTA pure FPLC system (Cytiva). A DAWN HELEOS II detector (Wyatt) and an Optilab TrEX differential refractometer (Wyatt) were used to estimate the molecular weight of $MRS2_{58-333}$ under various experimental conditions. The entire in-line FPLC/MALS system was housed in cold cabinet maintained at ~10°C. Data were obtained for four different protein concentrations: 0.45 mg/ml, 0.90 mg/ml, 2.5 mg/ml, and 5 mg/ml in 20 mM Tris (pH 8), 150 mM NaCl, and 1 mM DTT, using 100 µl injections of sample at each concentration. MALS molecular weights were determined in the accompanying ASTRA software (version 7.1.4; Wyatt) using Zimm plot analysis and a protein refractive index increment (dn/dc) = 0.185 L/g. Divalent cation containing experiments were performed by supplementing the running buffers and protein samples with 5 or 10 mM $MgCl_2$ and $CaCl_2$, as indicated.

### DLS

DLS measurements were made with a DynaPro NanoStar (Wyatt) instrument using a scattering angle of 90°. After centrifugation at 15,000g for 5 min at 4°C, 5 µl of supernatant was loaded into a JC-501 microcuvette, and measurements were collected as 10 consecutive acquisition scans with each acquisition being an average of 5 s. $MRS2_{58-333}$ protein samples were assessed at 1.25 mg/ml in 20 mM Tris (pH 8), 150 mM NaCl, and 1 mM DTT in the absence or presence of 5 mM $MgCl_2$, $CaCl_2$, or $CoCl_2$. Similarly, $MRS2_{58-443}$ protein samples were assessed at 0.5 mg/ml in 20 mM Tris (pH 8), 150 mM NaCl, 10 mM CHAPS, and 1 mM DTT, in the absence or presence of 5 mM $MgCl_2$, $CaCl_2$, or $CoCl_2$. For both proteins, data were acquired at 20 and 37°C, as indicated. All autocorrelation functions were deconvoluted using the regularization algorithm to extract the polydisperse distribution of hydrodynamic radii ($R_h$) and cumulants fit for monodisperse weight-averaged $R_h$ using the accompanying Dynamics software (version 7.8.1.3; Wyatt).

### Intrinsic fluorescence measurements for cation binding

A Cary Eclipse spectrofluorimeter (Agilent/Varian) was used to acquire intrinsic fluorescence emission spectra. Spectra were acquired for 0.1 mg/ml $MRS2_{58-333}$ in 20 mM Tris (pH 8), 150 mM NaCl, and 1 mM DTT, using a 600-µl quartz cuvette. The fluorescence emission intensities were recorded at 22.5°C from 300 to 450 nm, using a 1 nm data pitch and an excitation wavelength of 280 nm. Excitation and emission slit widths were set to 5 and 10 nm, respectively, and the photomultiplier tube detector was set to 650 V. Emission spectra were obtained before and after supplementation with increasing concentrations of $CaCl_2$, $MgCl_2$, or $CoCl_2$, added directly to the cuvette. A total of 15 emission spectra were acquired with increasing concentrations of divalent cation between 0 and 5 mM. Spectral intensities at 330 nm were corrected for the dilution associated with the volume change upon each addition to the cuvette, and resultant curves were fit to a one site binding model that takes into account protein concentration using R (version 4.2.1) to extract apparent equilibrium dissociation constants ($K_d$).

## Extrinsic 8-anilinonapthalene-1-sulfonic acid fluorescence

Extrinsic ANS fluorescence measurements were performed using a Cary Eclipse spectrofluorometer (Agilent/Varian). Spectra were acquired at 15°C for 30 µM MRS2$_{58-333}$ in 20 mM Tris (pH 8), 150 mM NaCl, 1 mM DTT, and 0.05 mM ANS, using a 600-µl quartz cuvette. The excitation wavelength was set to 372 nm, and the extrinsic ANS fluorescence emission spectra were acquired from 400 to 600 nm, with the photomultiplier tube detector set at 750 V. Excitation and emission slit widths were set to 10 and 20 nm, respectively, for all ANS experiments. To monitor divalent cation-induced changes in exposed hydrophobicity of MRS2$_{58-333}$, 5 mM CaCl$_2$ or 5 mM MgCl$_2$ was added directly into the cuvette. Negligible effects of these cations on free ANS fluorescence were confirmed by acquiring similar spectra in the absence of protein.

## Far-UV CD spectroscopy

Far-UV CD spectra were acquired using a Jasco J-810 CD spectrometer with electronic Peltier temperature regulator (Jasco). Each spectrum was taken as an average of 3 accumulations, recorded at 37°C using a 1-mm path length quartz cuvette in 1-nm increments, 8-s averaging time, and 1 nm bandwidth. To eliminate technical variability in magnitude signals, after acquiring divalent cation-free spectra, 5 mM MgCl$_2$ was added to the same samples, and spectra were re-acquired.

Thermal melts were recorded using a 1-mm path length quartz cuvette by monitoring the change in the CD signal at 222 nm from 20–95°C. A scan rate of 1°C min$^{-1}$, 1 nm bandwidth, and 8-s averaging time was used during data acquisition. Mg$^{2+}$-free and Mg$^{2+}$-supplemented data were fit using a Boltzmann sigmoidal equation to estimate the midpoint of temperature denaturation ($T_m$) using R (version 4.2.1).

## Mitochondrial Mg$^{2+}$ uptake experiments using Mag-Green

HeLa cells were cultured in DMEM with high glucose (Wisent), 10% (vol/vol) FBS (Sigma-Aldrich), 100 µg/ml penicillin, and 100 U/ml streptomycin (Wisent) at 37°C in a 5% CO$_2$, 95% (vol/vol) air mixture. Cells cultured in 35-mm dishes were transfected with PolyJet transfection reagent (SignaGen) according to the manufacturer guidelines. After ~12 h, cells were incubated with 0.725 µM of the Mg$^{2+}$ indicator Mag-Green for 30 min at 37°C. Cells were subsequently washed in divalent cation-free PBS, pH 7.4, and suspended in 2 ml of IB composed of 20 mM HEPES (pH 7), 130 mM KCl, 2 mM KH$_2$PO$_4$, 10 mM NaCl, 5 mM succinate, 5 mM malate, and 1 mM pyruvate. A 20% (vol/vol) cell suspension in IB was created in a quartz cuvette. Mag-Green fluorescence was monitored using a PTI QuantMaster spectrofluorimeter (Horiba) equipped with electronic temperature control using excitation and emission wavelengths of 506 and 531 nm, respectively, and excitation and emission slit widths of 2.5 and 2.5 nm, respectively. After a 30-s Mag-Green baseline fluorescence measurement, 2 mM EDTA plus 5 µM digitonin was added to permeabilize the PM. After 300 s, 3 mM MgCl$_2$ (or 10–30 mM NaCl for osmolarity controls) was added to the cuvette and the Mag-Green signal was measured for 600 s. The first three intensity values recorded immediately after the cation addback

were not included in any trace because of potential mixing and light artefact contributions. Mitochondrial Mg$^{2+}$ uptake was correlated with the clearance of Mg$^{2+}$, taken as the decrease in Mag-Green fluorescence after re-introduction of Mg$^{2+}$ into the system, as previously done (Daw et al, 2020). The rates of Mag-Green fluorescence decrease were extracted by fitting the traces after the Mg$^{2+}$ addback to a single exponential decay in R (version 4.2.1).

## Mito-Mag-FRET measurements in primary murine hepatocytes

Primary murine hepatocytes isolated from WT and Mrs2$^{-/-}$ (Daw et al, 2020) mice grown on 25-mm collagen-coated glass coverslips were transfected with empty vector, human MRS2-mRFP (Hu-MRS2-mRFP), or Hu-MRS2 D216K/D220K-mRFP plasmids. 24 h post-transfection, hepatocytes were transduced with adenoviral mito-Mag-FRET (Daw et al, 2020) (20 MOI) for an additional 48 h. The subcellular localizations of ectopically expressing MRS2 and mito-Mag-FRET were visualized using a Leica SP8 confocal microscope. The cells were excited using the 405-nm laser line, and the emission was collected using the hybrid detector (HyD). The cerulean channel, 460–490 nm, and citrine channel, 510–550 nm, served to detect the emissions from the fluorescence resonance energy transfer (FRET). FRET emissions were acquired following donor and acceptor excitation sequences. Selected region of interests (ROIs) were drawn, and the acquired sequences were background corrected for acceptor cross excitation cross-talk, acceptor cross excitation, and FRET cross-talk ($\alpha$ = A/C; $\gamma$ = B/C; $\delta$ = A/B). Time-lapse imaging was performed using the above-described acquisition mode, and the corresponding FRET efficiencies were analyzed. Selective ROIs focused on mitochondrial-targeted mito-Mag-FRET sensor signals, and the captured FRET sensitized emissions (relative FRET$_{SE}$) were plotted.

To compare Hu-MRS2-mRFP and Hu-MRS2 D216/KD220K-mRFP protein abundance, the pixel intensities of mRFP signals were evaluated by selecting multiple ROIs for the following conditions: WT + Hu-MRS2-mRFP, WT + Hu-MRS2$^{D216K/D220K}$-mRFP, Mrs2$^{-/-}$ + Hu-MRS2-mRFP, and Mrs2$^{-/-}$ + Hu-MRS2$^{D216K/D220K}$-mRFP.

## Structure modeling and visualization

The human MRS2 (UniProt accession Q9HD23) homopentamer model was generated using AlphaFold-Multimer (v2.2.0) (Evans et al, 2022 Preprint) on the Shared Hierarchical Academic Research Computing Network (SHARCNET:www.sharcnet.ca) of the Compute Canada/Digital Research Alliance of Canada. A total of 25 predictions were made (five seeds per model), using a maximum template date of 2022-11-01 and all available genetic databases. The highest confidence homopentamer model based on predicted LDDT was taken for visualization and analysis. All structure images were generated using PyMOL (Version 2.4; Schrödinger, LLC.).

## Statistics

Unpaired $t$ test was used when comparing two independent groups, paired $t$ test was used when comparing outcomes of the same group before and after treatment, and one-way ANOVA

after Tukey's post hoc test was used for multiple means comparisons between three or more groups. All nonlinear regression fitting and statistical analyses were done in GraphPad Prism (4.03) or R (4.2.1).

## Data Availability

The AlphaFold-Multimer generated human MRS2 homopentamer coordinates have been included as Source Data, and all other data are available from the corresponding author upon request.

## Supplementary Information

## Acknowledgements

This work was supported by a Canadian Institutes of Health Research Project Scheme Grant 438225 (to PB Stathopulos) and National Institutes of Health (R01GM109882, R01HL086699, R01HL142673, and R01GM135760) and DOD/DHP-CDMRP PR181598P-1 grants (to M Madesh).

### Author Contributions

S Uthayabalan: conceptualization, formal analysis, investigation, visualization, methodology, and writing—original draft.
N Vishnu: conceptualization, formal analysis, investigation, visualization, methodology, and writing—original draft.
M Madesh: conceptualization, resources, data curation, supervision, funding acquisition, validation, and writing—review and editing.
PB Stathopulos: conceptualization, resources, data curation, funding acquisition, validation, visualization, project administration, and writing—review and editing.

### Conflict of Interest Statement

The authors declare that they have no conflict of interest.

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
