## [Reviewer comments · Life Science Alliance]

Life Science Alliance

The human MRS2 magnesium binding domain is a regulatory feedback switch for channel activity.

Sukanthathulse Uthayabalan, Neelanjan Vishnu, Muniswamy Madesh, and Peter Stathopoulos

DOI: <https://doi.org/10.26508/lsa.202201742>

Corresponding author(s): Peter Stathopoulos, Western University

Review Timeline:

Submission Date:	2022-09-27
Editorial Decision:	2022-11-04
Revision Received:	2022-12-23
Editorial Decision:	2023-01-19
Revision Received:	2023-01-25
Accepted:	2023-01-26

Scientific Editor: Novella Guidi

Transaction Report:

November 4, 2022

Re: Life Science Alliance manuscript #LSA-2022-01742-T

Prof. Peter Stathopoulos
Western University
Western University
1151 Richmond Street
Med Sci Bldg, Rm 232
London, Ontario N6A5C1
Canada

Dear Dr. Stathopoulos,

Thank you for submitting your manuscript entitled "The human MRS2 magnesium binding domain is a regulatory feedback switch for channel activity." to Life Science Alliance. The manuscript was assessed by expert reviewers, whose comments are appended to this letter. We invite you to submit a revised manuscript addressing the Reviewer comments.

Thank you for this interesting contribution to Life Science Alliance. We are looking forward to receiving your revised manuscript.

Sincerely,

B. MANUSCRIPT ORGANIZATION AND FORMATTING:

Reviewer #1 (Comments to the Authors (Required)):

This study investigates the role of the magnesium binding domain of the human MRS2. This channel is responsible for the physiological accumulation of magnesium into the mitochondrial matrix and the proper bioenergetic functioning of the organelle. THE MRS2 contains two transmembrane domains that constitute the pore for Mg entry, but the role of the amino terminus domain in the mitochondrial matrix has remains elusive to these days. IN the present study, by using a combination of pinpointed residue mutations and a variety of biophysical and computational approaches, the authors have been able to demonstrate that the N-terminus of the protein acts as a Mg binding domain and regulates in a feedback switch manner the activity of the channel. Further, the authors were able to determine that Mg and Ca²⁺ ions suppress lower and higher order oligomerization of the MRS2 N-terminus domain, whereas cobalt has no effect. Disruption of the N-terminus Mg binding ability results in potentiation of Mg accumulation in the mitochondrial matrix, while selective residue mutations largely decrease Mg binding affinity and prevent Mg²⁺-induced changes in the secondary, tertiary and quaternary structures. The conclusion of the authors that the N-terminus of MRS2 functions as a negative feedback autoregulator for Mg accumulation in the mitochondrial matrix appears to be largely supported by the data reported here.

The study appears to be properly conducted and the conclusions are supported by the data reported in the manuscript.

Reviewer #2 (Comments to the Authors (Required)):

Comments:

The human MRS2 magnesium binding domain is a regulatory feedback switch for channel 2 activity - LSA-2022-01742-T

Uthayabalan et al have performed a comprehensive study on the composition, its regulation and the function of the NTD domain of MRS2. Using in silico and in vitro data, they show interestingly how certain residues within the NTD domain are required for MRS2 activity. This is particularly interesting as the gating mechanism of MRS2 is not well understood at the moment. The study would be of great value in the field. However, there are a few aspects that should be discussed within the manuscript:

Major comments:

1. You have investigated the effects of Mg²⁺ and Ca²⁺ on the assembly of the homodimers of the NTD-MRS2 proteins. Here, you've shown that Mg²⁺ has a quite profound effect on the assembly. In figure 5, you further describe the binding of the cations to the MRS2 proteins. Here, you demonstrate that the changes in concentration Ca²⁺ does not profoundly change the affinity and determined that the K_d is {plus minus} 1 mmol/L for Ca²⁺. What does this mean physiologically for the MRS2 protein? The mitochondrial concentration of Ca²⁺ is estimated at 50-500 nmol/L (<https://doi.org/10.3892/ijo.2019.4696>), suggesting that there would be little Ca²⁺ binding to the MRS2 protein in mitochondria, as I understand it. Does Ca²⁺ play a crucial role in MRS2 functioning/regulation?
2. Based on the structure of Pfoh et al, you identify that DALVD residues, specifically the Asp residues are important for the Mg²⁺ orientation. Does your mutagenesis in general affect protein structure? And please specify why you've chosen to change both residues simultaneously, and not one at a time.
3. Please explain why in figures FigS3A experiments have been performed at 20oC. You have investigated thermostability in silico/vitro in fig8. From a biophysical point of view, I understand why this would be interesting. However, using living cells at 20oC would greatly affect many physiological processes, including ion transport. In addition, why have these figures been put in the supplementary as you've spent a full section on highlighting these results with their own heading. Please move to main figures.
4. Please show MagGreen experiments where differences observed in intensity is really Mg²⁺ specific. This is for two reasons 1) MagGreen probes are not ratiometric, so are prone to swelling of cells/organelles. 3 mM MgCl₂ addition would change the osmolarity with 10mOsm/L (relatively small, but still). In addition, it would be interesting to determine if Ca²⁺ for example, which you hypothesize to bind to Mrs2, has an effect in the activity of MRS2 and the mutants.
5. Can you exclude from the experiments in figure S3 that Mg²⁺ has been taken up by mitochondria? The measurements are

performed after using digitonin, which permeabilizes the cells, allowing potentially MagGreen to leak out the cells, causing the decrease in cytosolic [Mg]. On the other hand, increase could potentially be in other compartments (ER/Golgi?). Using cuvettes to measure MagGreen does not allow one to determine the re-localization of the Mg²⁺ ions in the cells. Is live cell imaging not possible for your approach?

6. Figure 9: The images look very nice, but have a low magnification, making it very difficult to assess for the reader whether the Mito-Mag-FRET and Mrs2 localize to the mitochondria. Please show a higher magnification and (as supplementary) figure some images with MRS2/Mag-FRET and a marker for mitochondria (TMRE, Mitotracker?)

7. Same comment as 4, particularly adding 10mM MgCl₂ will drastically increase osmolarity. Please show that changes in signal are not derived due to these changes. CaCl₂ could be interesting here, although could have effects on MRS2, so alternative would NMDG-Cl/NaCl?

Minor comments:

Line 51-52: That Mg²⁺ is stored in different organelles remains a bit of a controversial topic. Indeed, there is a lot of Mg²⁺ in mitochondria and ER. However, these organelles require a lot of Mg²⁺ for MgATP and protein synthesis, respectively. Could you please provide references whether Mg²⁺ is really stored? Otherwise please rephrase.

Line 64: Mutations in the Na⁺/Mg²⁺ exchanger; first of all, please specify which Na/Mg this is as there are multiple. In addition, although this mutation is associated to Parkinson disease, whether Mg is really involved in Parkinson development/progression has not been well studied. There are many other diseases whether cellular and systemic Mg²⁺ is affected (TRPM6/ TRPM7/ CNNM2/CNNM4). To illustrate your point, I would suggest referencing to one of these proteins.

Line 191-192: the DALVD residues are conserved in prokaryotic CorA and human MRS2, but not in yeast. Do you have a hypothesis where the residues are located in yeast MRS2 where it binds Mg²⁺ ions? Do more species not have this conserved DALVD sequence?

Figure 9: Would it be possible to co-express WT and the double mutants and perform the FRET experiments? In the MRSS KO cells, you overexpression with WT or mutant MRS2 allows you to study homopentamers, and their effects on MRS2 activity. But it would be interesting to see if heteropentamers would show similar results as the mutant homopentamer. You have tried this in 9D to some extent, but the overexpression will likely ensure boost levels of MRS2-RFP to such an extent that heteromers are maybe not formed anymore.

Line 307: "are similarly well-expressed" This is difficult to assess from microscopic images. Do you have a quantification either based on intensity of the microscopy images or Western blot analysis?

Reviewer #3 (Comments to the Authors (Required)):

This work by the group of Dr. Stathopoulos characterized human MRS2 Mg²⁺ transporter by a combination of biochemistry and cell biology-based assays. In the MRS2/CorA superfamily, whereas bacterial CorA proteins have been well-characterized by structural biology and biophysics, eukaryotic MRS2 proteins, in particular human MRS2, were poorly characterized. Therefore, this work would attract a wide range of interests for readers in the field. However, there is a series of technical problems, in particular with the sequence alignment and homology model they presented. Thus, while I recommend this manuscript for publication, the authors should extensively revise the manuscript to address the following concerns before publication.

1. Sequence alignment

The sequence alignment in Figure 1B is inappropriate, in particular, that for *T. maritima* CorA. Despite the low sequence identity between eukaryotic MRS2 and bacterial CorA, the authors included only one bacterial CorA sequence in the alignment, which caused serious errors in the sequence alignment result, leading to the misinterpretation in the comparison between the predicted structure of hMRS2 and experimental CorA structures and the speculation on the putative Mg²⁺ binding site (See below). The authors should include more bacterial CorA sequences to generate a new alignment, which would be more consistent with the AlphaFold-based predicted model. Actually, before preparing this review comment, I generated a new sequence alignment and AlphaFold2-based prediction model, which looked more reasonable.

2. Prediction structure of human MRS2

Page 16 "The AlphaFold (Jumper et al, 2021) model of human MRS2 places the M2 binding residues closer to the PM region of the NTD, inconsistent with the near β -sheet position elucidated in the CorA crystal and cryoEM structures (Fig. 10B). Thus, we created a homology model in Modeller (Webb & Sali, 2017), positioning the DALVD residues, conserved with CorA, in a structurally similar position as elucidated by the CorA crystal structures (Fig. 10C)."

As I mentioned above, since the original sequence alignment with MRS2 and *T. maritima* CorA in Figure 1B is inappropriate, the homology model by Modeller also includes the same problem. They should make a new sequence alignment with more bacterial CorA proteins and should use the AlphaFold2 based prediction structure of human MRS2. In addition, the authors should not use the monomeric form of the prediction structure for discussion. They should use the pentameric form since Mg²⁺ binding sites are located at the subunit interfaces.

3. Putative cytoplasmic Mg²⁺ binding site of human MRS2

Page 8 "We next attempted to pinpoint the residues involved in Mg²⁺ coordination using the CorA crystal structure (4EED.pdb)

(Pfoh et al., 2012) as a guide. Note that available yeast Mrs2 structures do not resolve any Mg²⁺ ions bound to the NTD. The CorA crystal structure shows that two Asp residues separated by three residues (i.e. DALVD) are involved in Mg²⁺ coordination at one site. Remarkably, the DALVD sequence is fully conserved in human MRS2 but not yeast (Fig. 1B).

If the authors generate a new sequence alignment with more bacterial CorA sequences, they would notice that Asp216 and Asp220 in human MRS2 do not correspond to Asp175 and Asp179 in *T. maritima* CorA, respectively. Instead, Asp216 and Asp220 residues correspond to Glu197 and Glu201 in *T. maritima* CorA, respectively, whereas Asp175 and Asp179 in *T. maritima* CorA correspond to Gln194 and Asn198 in human MRS2, respectively. Thus, The cytoplasmic Mg²⁺ binding in *T. maritima* CorA is not conserved in human MRS2. Overall, the authors should not mention Asp216 and Asp220 as potential Mg²⁺ binding residues in the revised manuscript.

In the AlphaFold based prediction model I generated, Glu243 and Asp247 face Asp305 and Glu312 in another subunit, respectively. In addition, these residues are well conserved among animal MRS2 proteins. Mg²⁺ binding sites are typically located at such cluster regions including conserved acid residues. So, if the authors still plan to discuss potential cytoplasmic Mg²⁺ binding sites of MRS2, the authors should test mutants of such residues. Without such extra experiments, the authors should not discuss a potential cytoplasmic Mg²⁺ binding site of MRS2.

4. Interpretation of D216A/D220 mutant

In the AlphaFold-based prediction model, Asp216 and Asp220 are located at the subunit interface and potentially interact with His232, Gln236 and/or Lys239. Therefore, while these residues would not be directly involved in Mg²⁺ binding, the mutations to Asp216 and Asp220 would still affect the pentamer formation of the full-length MRS2, which would affect the negative feedback mechanism of MRS2. In fact, the authors observed the upregulated Mg²⁺ transport in D216/D220A mutant, but this can be explained by the destabilization of pentamer formation of the full-length MRS2. The authors should revise the interpretation of this mutant in the revised manuscript if they plan to include the data on this mutant in the revised manuscript.

Our responses (red text) are shown directly below each specific comment by the reviewers (italics).

Reviewer #1 (Comments to the Authors (Required)):

This study investigates the role of the magnesium binding domain of the human MRS2. This channel is responsible for the physiological accumulation of magnesium into the mitochondrial matrix and the proper bioenergetic functioning of the organelle. The MRS2 contains two transmembrane domains that constitute the pore for Mg entry, but the role of the amino terminus domain in the mitochondrial matrix has remains elusive to these days. IN the present study, by using a compbination of pinpointed residue mutations and a variety of biophysical and computational approaches, the authors have been able to demonstrate that the N-terminus of the protein acts as a Mg binding domain and regulates in a feedback switch manner the activity of the channel. Further, the authors were able to determine that Mg and Ca²⁺ ions suppress lower and higher order oligomerizatin of the MRS2 N-terminus domain, whereas cobalt has no effect. Disruption of the N-terminus Mg binding ability results in potentiation of Mg accumulation in the mitochondrial matrix, while selective residue mutations largely decrease Mg binding affinity and prevent Mg=-induced changes in the secondary, tertiary and quaternary structures. The conclusion of the authors that the N-terminus of MRS2 functions as a negative feedback autoregulator for Mg accumulation in the mitochondrial matrix appears to be largely supported by the data reported here.

The study appears to be properly conducted and the conclusions are supported by the data reported in the manuscript.

We thank the reviewer for their time in reviewing our manuscript and for reinforcing the importance of the work.

Reviewer #2 (Comments to the Authors (Required)):

Comments:

The human MRS2 magnesium binding domain is a regulatory feedback switch for channel 2 activity - LSA-2022-01742-T

Uthayabalan et al have performed a comprehensive study on the composition, its regulation and the function of the NTD domain of MRS2. Using in silico and in vitro data, they show interestingly how certain residues within the NTD domain are required for MRS2 activity. This is particularly interesting as the gating mechanism of MRS2 is not well understood at the moment. The study would be of great value in the field. However, there are a few aspects that should be discussed within the manuscript:

We are grateful to the reviewer for their time in reviewing our manuscript and for appreciating the value added to the Mg²⁺ signaling research field. In response to the suggestions, we have incorporated new data and clarified the manuscript with text revisions.

Major comments:

1. You have investigated the effects of Mg²⁺ and Ca²⁺ on the assembly of the homodimers of the NTD-MRS2 proteins. Here, you've shown that Mg²⁺ has a quite profound effect on the assembly. In figure 5, you further describe the binding of the cations to the MRS2 proteins. Here, you demonstrate that the

changes in concentration Ca^{2+} does not profoundly change the affinity and determined that the K_d is {plus minus} 1 mmol/L for Ca^{2+} . What does this mean physiologically for the MRS2 protein? The mitochondrial concentration of Ca^{2+} is estimated at 50-500 nmol/L (<https://doi.org/10.3892/ijo.2019.4696>), suggesting that there would be little Ca^{2+} binding to the MRS2 protein in mitochondria, as I understand it. Does Ca^{2+} play a crucial role in MRS2 functioning/regulation?

Given the relatively weak Ca^{2+} binding affinity we estimated for the MRS2 NTD (i.e. $K_d \sim 1$ mM; see **Fig. 5B, Fig. 5E** and **Table S3**) and the mitochondrial Ca^{2+} concentration noted by the reviewer (i.e. up to 0.0005 mM) and others after cell stimulation (i.e. typically $\sim 0.010 - 0.1$ mM) (Fernandez-Sanz *et al*, 2019), it is unlikely Ca^{2+} binding to the NTD plays a crucial role in MRS2 regulation. Congruently, we previously showed that MCU knockout (KO) hepatocytes do not alter lactate stimulated mitochondrial Mg^{2+} uptake (Daw *et al*, 2020). One way we envision Ca^{2+} could regulate MRS2 via binding to the NTD would be for the domain to be oriented very close to a Ca^{2+} channel pore exit (Bauer, 2001; Chad & Eckert, 1984), where Ca^{2+} concentrations have been estimated in the \sim mM range; however, sterically, this close co-assembly with a Ca^{2+} channel is unlikely.

We have added these considerations to the **Discussion, lines 424-434**.

2. Based on the structure of Pfoh *et al*, you identify that DALVD residues, specifically the Asp residues are important for the Mg^{2+} orientation. Does your mutagenesis in general affect protein structure? And please specify why you've chosen to change both residues simultaneously, and not one at a time.

There are several lines of evidence indicating that the D216A/D220A double mutation minimally affects MRS2 protein structure. First, our CD measurements demonstrate that D216A/D220A does not change secondary structure compared to the WT MRS2-NTD (see **Fig 8A** and **8B**). Second, thermal stability of WT and mutant MRS2-NTD are identical (see **Fig. 8E** and **8F**). Third, the mutant and WT MRS2-NTDs show similar ANS binding and fluorescence intensities, indicating no change in tertiary structure (see **Fig 6**). Fourth, SEC-MALS and DLS show no difference in the quaternary structures of the mutant compared to WT (see **Fig. 2A** versus **Fig. 7A** and **Fig. 7B**, **Fig. 3A** versus **Fig. 7B**, **Table S1** and **Table S2**). Fifth, the Co^{2+} and Ca^{2+} K_d s are not altered by the mutation, suggesting no structural changes propagating to other NTD regions (see **Fig. 5** and **Table S3**).

We created the D216 and D220 double mutant as a single Asp side chain would be insufficient to coordinate the Mg^{2+} with the ~ 0.14 mM affinity we measured (see **Fig. 5** and **Table S3**). Additionally, the D216 and D220 would be very close in three-dimensional (3D) space in the context of an α -helix, in which most of the DALVD region is predicted to exist using AlphaFold2 (see **Fig. 11B**), also suggesting both Asp would coordinate the same Mg^{2+} ion. Similarly, the bacterial *T. maritima* CorA structure shows both side chains in the identical DALVD sequence coordinate the same Mg^{2+} ion (see **Fig. 11A**).

We have added these considerations to the **Results, lines 206-210**.

3. Please explain why in figures FigS3A experiments have been performed at 20oC. You have investigated thermostability in silico/vitro in fig8. From a biophysical point of view, I understand why this would be interesting. However, using living cells at 20oC would greatly affect many physiological processes, including ion transport. In addition, why have these figures been put in the supplementary as you've spent a full section on highlighting these results with their own heading. Please move to main figures.

We were motivated to evaluate uptake at 20 °C since previous data indicated MRS2 is less active at lower temperature (Daw *et al.*, 2020). However, our 20 °C data showed a higher variability compared to 37 °C, which could be related to the low temperature affecting multiple physiological processes (as indicated by the reviewer). Although Mg²⁺ clearance rates appeared slightly higher at 37 °C compared to 20 °C for the D216A/D220A double mutant, the WT-MRS2 and empty-vector control-transfected cells showed similar rates at both temperatures.

Thus, given the concerns of the reviewer with respect to the low temperature and the generally greater variability, we have removed the 20 °C data. Additionally, we have now integrated the 37 °C data as a main text figure, as suggested by the reviewer, and we have included fitted decays that match the representative responses shown above (see revised **Results**, lines 288-307 and new main text **Fig. 9**).

4. Please show MagGreen experiments where differences observed in intensity is really Mg²⁺ specific. This is for two reasons 1) MagGreen probes are not ratiometric, so are prone to swelling of cells/organelles. 3 mM MgCl₂ addition would change the osmolarity with 10mOsm/L (relatively small, but still). In addition, it would be interesting to determine if Ca²⁺ for example, which you hypothesize to bind to Mrs2, has an effect in the activity of MRS2 and the mutants.

In our permeabilized experiments, cells were suspended in an intracellular buffer (20 mM HEPES, pH 7.2, 130 mM KCl, 2 mM KH₂PO₄, 10 mM NaCl, 5 mM succinate, 5 mM malate and 1 mM pyruvate), which keeps mitochondrial membrane potential intact. This approach has been used by other laboratories including ours to measure mitochondrial Ca²⁺ and Mg²⁺ uptake upon extramitochondrial divalent cation addition (Daw *et al.*, 2020; Mallilankaraman *et al.*, 2012). Nevertheless, to confirm that our responses using the Mag-Green experiments were not affected by osmolarity changes, we repeated our 37 °C experimental protocol and added 10 and 30 mM NaCl to the system to recapitulate the osmolarity changes caused by our 3 mM and 10 mM MgCl₂ additions in **Fig. 9** and **Fig. 10**, respectively. The Mag-Green fluorescence signal did not change after the addition of the NaCl, suggesting that osmolarity changes likely did not contribute to any detectable differences using MgCl₂.

Given the weak MRS2-NTD Ca²⁺ binding affinity we measured (*i.e.* K_d ~1 mM; see **Fig. 5** and **Table S3**), we anticipate aberrantly high matrix Ca²⁺ concentrations would be needed to effect MRS2 activity. For example, 0.10 mM Ca²⁺ would occupy < 10% of the MRS2 Ca²⁺ binding sites, and thus, matrix Ca²⁺ concentrations needed to occupy most of the binding sites, where an effect could easily be detected, would collapse the membrane potential. Thus, assessing how the weak Ca²⁺ binding we characterized *in vitro* affects MRS2 activity *in cellulo* would be technically challenging.

We have included our osmolarity assessment as new **Fig. S5** and a discussion of the technical challenges of detecting Ca²⁺-dependent regulation of MRS2 on **Discussion**, lines 424-434.

5. Can you exclude from the experiments in figure S3 that Mg²⁺ has been taken up by mitochondria? The measurements are performed after using digitonin, which permeabilizes the cells, allowing potentially MagGreen to leak out the cells, causing the decrease in cytosolic [Mg]. On the other hand, increase could potentially be in other compartments (ER/Golgi?). Using cuvettes to measure MagGreen does not allow one to determine the re-localization of the Mg²⁺ ions in the cells. Is live cell imaging not possible for your approach?

We previously showed that the apparent amount of extramitochondrial Mg²⁺ cleared by permeabilized cells, as detected using the Mag-Green approach, is comparable to total mitochondrial Mg²⁺ release upon subsequent addition of a membrane potential uncoupler [please see Fig. 3F and 3G of (Daw *et al.*, 2020)]. Nevertheless, we agree that Mag-Green leakage and Mg²⁺ uptake in other compartments such as ER/Golgi may have contributed to our signals. Therefore, as suggested, we applied an intact live cell imaging approach for direct mitochondrial Mg²⁺ uptake assessment to validate the permeabilized cell system data (see **Fig. 10**).

6. *Figure 9: The images look very nice, but have a low magnification, making it very difficult to assess for the reader whether the Mito-Mag-FRET and Mrs2 localize to the mitochondria. Please show a higher magnification and (as supplementary) figure some images with MRS2/Mag-FRET and a marker for mitochondria (TMRE, Mitotracker?)*

We previously showed that our mito-Mag-FRET sensor properly localizes to mitochondria, reporting reciprocal lactate-induced Mg²⁺ responses compared to an ER-targeted and retained Mag-FRET sensor (Daw *et al.*, 2020). We also showed that adenoviral-mediated and overexpressed murine Mrs2-mRFP under the control of a CMV promoter (similar to the one used in the present manuscript) co-localizes with the mitochondrial marker dihydorhodamine 123 (Daw *et al.*, 2020). Here, human MRS2-mRFP and the mito-Mag-FRET sensor show the same perinuclear, filamentous and interconnected distribution (**Fig. 10A, 10B** and new **Fig. S6A, S6B**), also suggesting both are properly localized in mitochondria. In addition, we compared the intensity profiles of the mito-Mag-FRET and human MRS2-mRFP fluorescence expressed in WT hepatocytes. The pixel intensity profiles of the mito-Mag-FRET citrine and human MRS2-mRFP (WT and mutant) signals showed co-incident peak maxima, further reinforcing the co-localization of the proteins in the mitochondria (see new **Fig. S7**).

To keep the manuscript size manageable for review (as per journal guidelines), the highest resolution images showing localization were not included in the original submission. Further, the resolution of these images may have been further reduced by the manuscript submission and merging system. In our revisions, we have included larger/higher resolution images with more discernable subcellular detail, showing similar mito-Mag-FRET (*i.e.* used as a mitochondrial marker) and MRS2-mRFP localization (see new **Fig. S6A** and **S6B**). We have also included the line scan analyses showing co-incident profiles for mito-Mag-FRET and human MRS2-mRFP fluorescence intensity (see new **Fig. S7** and **Results, lines 324-327**), and we have made note in the **Results, lines 315-320** that the mito-Mag-FRET and adenoviral-mediated/ectopically expressed Mrs2-mRFP proteins have been previously shown to properly localize to the mitochondria (Daw *et al.*, 2020).

7. *Same comment as 4, particularly adding 10mM MgCl₂ will drastically increase osmolarity. Please show that changes in signal are not derived due to these changes. CaCl₂ could be interesting here, although could have effects on MRS2, so alternative would NMDG-Cl/NaCl?*

As mentioned above, we collected new data showing that 30 mM NaCl, which has comparable osmolarity to 10 mM MgCl₂, does not alter the Mag-Green signal in our permeabilized cell system. More specific to the intact/live cell experiments mentioned here, our Mrs2(-/-) KO hepatocyte experiments were used as a negative control. The Mrs2 KO hepatocytes showed only a small ~4% change in the relative FRET sensitized emissions after 10 mM MgCl₂ addition, in contrast to the WT- and mutant-overexpressing groups, which showed ~13% and 30% changes, respectively, after the same 10 mM MgCl₂ bolus. If the signal changes were derived from osmolarity differences, then all groups would show

similar responses; moreover, if MRS2-mRFP co-localized to mitochondria with mito-Mag-FRET somehow affected the osmolarity responses, then both of these groups (*i.e.* WT and mutant MRS2-mRFP) would be expected to have similar responses, which are inconsistent with our observations. Note that intact cells bathed in HBSS (nominally free of CaCl₂ and MgCl₂) were used in these live cell assays, where the HBSS has an osmolarity of ~300 mOsm/L; therefore, addition of 10 mM MgCl₂ would change the osmolarity of the bathing medium by a modest ~10 %.

We have added the 30 mM NaCl data as a new **Fig. S5B** and these considerations in the **Discussion, lines 450-452**.

Minor comments:

Line 51-52: That Mg²⁺ is stored in different organelles remains a bit of a controversial topic. Indeed, there is a lot of Mg²⁺ in mitochondria and ER. However, these organelles require a lot of Mg²⁺ for MgATP and protein synthesis, respectively. Could you please provide references whether Mg²⁺ is really stored? Otherwise please rephrase.

As suggested, we have rephrased original manuscript lines 51-52 to state “Remarkably, Mg²⁺ can be mobilized from the endoplasmic reticulum (ER) in response to ligands such as L-lactate, moving into the mitochondria and dramatically modifying metabolism (Daw et al, 2020)” (see revised manuscript **Introduction, lines 52-53**).

Line 64: Mutations in the Na⁺/Mg²⁺ exchanger; first of all, please specify which Na/Mg this is as there are multiple. In addition, although this mutation is associated to Parkinson disease, whether Mg is really involved in Parkinson development/progression has not been well studied. . There are many other diseases whether cellular and systemic Mg²⁺ is affected (TRPM6/ TRPM7/ CNNM2/CNNM4). To illustrate your point, I would suggest referencing to one of these proteins.

As suggested, we have specified that the A350V mutation in the Na⁺/Mg²⁺ exchanger SLC41A1 results in enhanced Na⁺-dependent Mg²⁺ efflux by ~70% in HEK293 cells (Kolisek et al, 2013); further, we mention that low Mg²⁺ intake in rats leads to loss of dopaminergic neurons (Oyanagi et al, 2006) and in humans is linked with an increased risk of idiopathic Parkinson’s disease (Aden et al, 2011). Finally, we state that decreased free cytosolic Mg²⁺ has been measured in the occipital lobes of Parkinson’s patients (Barbiroli et al, 1999). We also now provide a secondary example in TRPM6, where mutations result in Mg²⁺ malabsorption and renal wasting leading to hypomagnesemia and hypocalcemia; affected individuals show seizures and muscle spasms during infancy (Schaffers et al, 2018; Schlingmann et al, 2002).

We have added these additional details and example in the **Introduction, lines 67-75**.

Line 191-192: the DALVD residues are conserved in prokaryotic CorA and human MRS2, but not in yeast. Do you have a hypothesis where the residues are located in yeast MRS2 where it binds Mg²⁺ ions? Do more species not have this conserved DALVD sequence?

Since the sequence conservation between bacterial CorA and eukaryotic MRS2 including yeast is low, we are not confident in the multiple sequence alignments. This uncertainty is highlighted by the different alignment patterns output by T-COFFEE (Notredame et al, 2000) and Clustal Omega (Sievers et al, 2011), two of the most commonly used sequence alignment programs, using the same input sequences (see

new **Fig. S1** and **S2**). Thus, to assess more reliably whether the DALVD Mg²⁺ M2 binding site of *T. maritima* CorA is conserved in yeast Mrs2, we performed a 3D structural alignment using the TM-align server (Zhang & Skolnick, 2005). A superposition of the *T. maritima* CorA (4EED.pdb) and *S. cerevisiae* Mrs2 (3RKG.pdb) crystal structures showed that the bacterial CorA DALVD sequence structurally aligns with an INVMS sequence in *S. cerevisiae* Mrs2 (*i.e.* the Mg²⁺ binding site is not conserved).

However, since AlphaFold2 orients the human DALVD sequence closer to the membrane domains (but within the analogous helix) compared to the bacterial CorA DALVD, we also superposed the human MRS2 AlphaFold2 model with the yeast Mrs2 crystal structure, finding the human DALVD sequence is predicted to structurally align with a yeast DLENE sequence. In this scenario, the two Asp in DALVD are apparently conserved as Asp and Glu in the yeast DLENE structure. Ultimately, experimentally determined human MRS2 and yeast Mrs2 structures in the presence of Mg²⁺ are needed to know precisely how Mg²⁺ coordination differs (or is similar) among these species.

We have included the structural alignment of *T. maritima* CorA and *S. cerevisiae* Mrs2 as well as the AlphaFold2 human MRS2 model and *S. cerevisiae* Mrs2 as new **Fig. S8A** and **Fig. S8B**, clarifying that the human Mg²⁺ binding site at DALVD is possibly conserved in the lower eukaryote but noting that high resolution structural data is needed to precisely answer this question (see **Discussion, lines 382-393**).

Figure 9: Would it be possible to co-express WT and the double mutants and perform the FRET experiments? In the MRSS KO cells, you overexpression with WT or mutant MRS2 allows you to study homopentamers, and their effects on MRS2 activity. But it would be interesting to see if heteropentamers would show similar results as the mutant homopentamer. You have tried this in 9D to some extent, but the overexpression will likely ensure boost levels of MRS2-RFP to such an extent that heteromers are maybe not formed anymore.

While the human MRS2 structure probably exists as a pentamer based on our knowledge of bacterial CorA, the true functional stoichiometry remains unresolved. Discerning whether heteromers and/or homomers contribute to increased Mg²⁺ uptake by co-overexpressing wild-type and mutant MRS2 would be difficult since WT or mutant MRS2 expression significantly restores Mg²⁺ uptake in the KO murine hepatocytes (**Fig. 10E**). Additionally, co-expression would not precisely control stoichiometric ratios, and the channel could be differentially regulated by integration of one or more mutant protomers. An alternative approach would be to create concatemers enforcing specific wild-type:mutant protomer ratios. However, since the precise stoichiometry of functional human MRS2 remains unknown, performing a concatemer experiment is premature. We appreciate this is an interesting and important question, but hope to pursue this question in follow up studies once functional stoichiometry of human MRS2 is firmly established.

Line 307: "are similarly well-expressed" This is difficult to assess from microscopic images. Do you have a quantification either based on intensity of the microscopy images or Western blot analysis?

As suggested, we have performed an intensity quantification of our image micrographs overexpressing wild-type and mutant MRS2-mRFP, finding no difference in the intensities among the four groups where we evaluated mitochondrial Mg²⁺ uptake. We have included this intensity analysis as new **Fig. S6C, Results, lines 334-337**.

Reviewer #3 (Comments to the Authors (Required)):

This work by the group of Dr. Stathopoulos characterized human MRS2 Mg²⁺ transporter by a combination of biochemistry and cell biology-based assays. In the MRS2/CorA superfamily, whereas bacterial CorA proteins have been well-characterized by structural biology and biophysics, eukaryotic MRS2 proteins, in particular human MRS2, were poorly characterized. Therefore, this work would attract a wide range of interests for readers in the field. However, there is a series of technical problems, in particular with the sequence alignment and homology model they presented. Thus, while I recommend this manuscript for publication, the authors should extensively revise the manuscript to address the following concerns before publication.

We thank the reviewer for taking the time to review our manuscript, appreciating the importance of the work and suggestions to improve our Fig. 1B sequence alignment and original submission Fig. 10 high-resolution model. In response, we have revised the alignment, Fig. 10 model (now revised Fig. 11) and associated discussion points to be in-line with the reviewer suggestions.

1. Sequence alignment

*The sequence alignment in Figure 1B is inappropriate, in particular, that for *T. maritima* CorA. Despite the low sequence identity between eukaryotic MRS2 and bacterial CorA, the authors included only one bacterial CorA sequence in the alignment, which caused serious errors in the sequence alignment result, leading to the misinterpretation in the comparison between the predicted structure of hMRS2 and experimental CorA structures and the speculation on the putative Mg²⁺ binding site (See below). The authors should include more bacterial CorA sequences to generate a new alignment, which would be more consistent with the AlphaFold-based predicted model.*

Our sequence alignment in **Fig. 1B** was performed using T-COFFEE [(Notredame *et al.*, 2000); 8,100 citations in Google Scholar]. Addition of 13 more bacterial CorA sequences does not change the alignment of the human DALVD sequence with the identical DALVD sequence of *T. maritima* using T-COFFEE. In contrast, using Clustal Omega [(Sievers *et al.*, 2011); 13,100 citations in Google Scholar], the human DALVD sequence aligns with a different part of the *T. maritima* sequence, as the reviewer points out. Note that the 13 bacterial species included are the complete list curated in the NCBI Landmark Database spanning a diverse, non-redundant and wide taxonomic range.

Thus, as suggested, we have added the multiple sequence alignments output by T-COFFEE and Clustal Omega that include more bacterial CorA sequences, highlighting the algorithm-dependent differences in alignment due to the poor sequence conservation between human MRS2 and bacterial CorA. (see **Results lines 201-205, Fig. 1B legend lines 822-825** and new **Fig. S1** and **Fig. S2**).

Actually, before preparing this review comment, I generated a new sequence alignment and AlphaFold2-based prediction model, which looked more reasonable.

Our original manuscript showed the AlphaFold2 predicted structure of human MRS2 in Fig. 10B, downloaded directly from the AlphaFold Protein Structure Database (<https://alphafold.ebi.ac.uk/entry/Q9HD23>). We also generated our own human MRS2 model using a local implementation of AlphaFold2 (v2.2), which appears identical to the downloaded prediction through the structured regions. If the reviewer has generated a model that “looked more reasonable” than the one shown in our original Fig. 10B (**Fig. 11B** in the revised submission), there may be an error in

their implementation or usage of AlphaFold2, as we feel it is unlikely that the deposited MRS2 model and a similar but independently generated AlphaFold2 model are both different than the version generated by the reviewer.

2. Prediction structure of human MRS2

Page 16 "The AlphaFold (Jumper et al, 2021) model of human MRS2 places the M2 binding residues closer to the PM region of the NTD, inconsistent with the near β -sheet position elucidated in the CorA crystal and cryoEM structures (Fig. 10B). Thus, we created a homology model in Modeller (Webb & Sali, 2017), positioning the DALVD residues, conserved with CorA, in a structurally similar position as elucidated by the CorA crystal structures (Fig. 10C)."

As I mentioned above, since the original sequence alignment with MRS2 and *T. maritima* CorA in Figure 1B is inappropriate, the homology model by Modeller also includes the same problem. They should make a new sequence alignment with more bacterial CorA proteins and should use the AlphaFold2 based prediction structure of human MRS2. In addition, the authors should not use the monomeric form of the prediction structure for discussion. They should use the pentameric form since Mg²⁺ binding sites are located at the subunit interfaces.

Indeed, we showed the AlphaFold2 structure in our original Fig. 10B, which must have been overlooked by the reviewer. We note that the AlphaFold2 prediction confidence for the Asp216 and Asp220 residues (the focus of our experimental work) of human MRS2 is not considered very high (i.e. pLDDT < 90). Additionally, the functional stoichiometry of the human MRS2 channel remains unknown, although we agree that a homopentamer is probable. Nevertheless, we also generated a pentameric structure using AlphaFold2, as suggested. Just as for the monomer prediction, the Asp216 and Asp220 residues and entire DALVD sequence show pLDDT scores of < 90. Therefore, while the backbone conformations predicted for these residues are probably correct, the predicted orientations of the side chains are not reliable, and an experimentally determined human MRS2 structure is needed to firmly establish the precise functional stoichiometry of the channel and the positions of the Asp216 and Asp220 side chains with respect to the assembled channel.

As suggested, the AlphaFold2 pentameric model is now shown as new **Fig. 11B** and new **Fig. 11C**. Additionally, we have added text in the **Fig. 11 legend, lines 1009-1014**, describing the AlphaFold2 prediction confidence for DALVD and adjacent loop sequence and reiterating that an experimentally determined human MRS2 high-resolution structure is needed to firmly establish the functional stoichiometry of the channel and the positions of the Asp216 and Asp220 side chains with respect to the assembled channel. Also, given the concerns regarding the alignment, we have eliminated the Modeller model from the manuscript and solely focused on the AlphaFold2 prediction.

3. Putative cytoplasmic Mg²⁺ binding site of human MRS2

Page 8 "We next attempted to pinpoint the residues involved in Mg²⁺ coordination using the CorA crystal structure (4EED.pdb) (Pfoh et al., 2012) as a guide. Note that available yeast Mrs2 structures do not resolve any Mg²⁺ ions bound to the NTD. The CorA crystal structure shows that two Asp residues separated by three residues (i.e. DALVD) are involved in Mg²⁺ coordination at one site. Remarkably, the DALVD sequence is fully conserved in human MRS2 but not yeast (Fig. 1B).

We thank the reviewer for suggesting we look closer at the alignment. Since the T-COFFEE and Clustal Omega output different alignments (see our response to reviewer #3, point #1 above), we have revised the last sentence to read “Remarkably, human MRS2 contains an identical DALVD sequence, which could similarly coordinate Mg²⁺” (see **Results, lines 200-201**).

If the authors generate a new sequence alignment with more bacterial CorA sequences, they would notice that Asp216 and Asp220 in human MRS2 do not correspond to Asp175 and Asp179 in T. maritima CorA, respectively. Instead, Asp216 and Asp220 residues correspond to Glu197 and Glu201 in T. maritima CorA, respectively, whereas Asp175 and Asp179 in T. maritima CorA correspond to Gln194 and Asn198 in human MRS2, respectively. Thus, The cytoplasmic Mg²⁺ binding in T. maritima CorA is not conserved in human MRS2. Overall, the authors should not mention Asp216 and Asp220 as potential Mg²⁺ binding residues in the revised manuscript.

In the AlphaFold based prediction model I generated, Glu243 and Asp247 face Asp305 and Glu312 in another subunit, respectively. In addition, these residues are well conserved among animal MRS2 proteins. Mg²⁺ binding sites are typically located at such cluster regions including conserved acid residues. So, if the authors still plan to discuss potential cytoplasmic Mg²⁺ binding sites of MRS2, the authors should test mutants of such residues. Without such extra experiments, the authors should not discuss a potential cytoplasmic Mg²⁺ binding site of MRS2.

The suggestion that we “should not mention Asp216 and Asp220 as a Mg²⁺ binding site” is unjustified as it overlooks the experimental data we collected showing Asp216/Asp220-dependent Mg²⁺ binding and structural sensitivity (*i.e.* see **Fig. 5, Fig. 6, Fig. 7, Fig. 8, Fig. S3, Fig. S4, Table S1, Table S2 and Table S3**). These experimental observations are the focus of the manuscript and are independent of the alignment and model predictions focused on by this reviewer. We stress that the models generated in **Fig. 11** are only predictions and discussion points used to speculate how Mg²⁺ binding at the Asp216/Asp220 site could result channel inhibition.

Our data showed that Asp216Ala/Asp220Ala mutations disrupt Mg²⁺-binding (see **Fig. 5A and Fig. 5D; Table S3**) and the associated biophysical and structural effects without altering the protein secondary structure (see **Fig 8A and 8B**), stability (see **Fig. 8E and Fig. 8F**), tertiary structure (see **Fig 6**), quaternary structures and higher order assemblies (see **Fig. 2A versus Fig. 7A and Fig. 7B, Fig. 3A versus Fig. 7C, Table S1 and Table S2**). Nevertheless, we thank the reviewer for pointing out this additional divalent cation binding site possibility. Thus, we have indicated that Glu243, Asp247, Asp305 and Glu312 may be involved in forming additional cation binding site(s), besides the binding site we have experimentally identified and validated (see new **Fig. 11C, Discussion, lines 404-408**).

4. Interpretation of D216A/D220 mutant

In the AlphaFold-based prediction model, Asp216 and Asp220 are located at the subunit interface and potentially interact with His232, Gln236 and/or Lys239. Therefore, while these residues would not be directly involved in Mg²⁺ binding, the mutations to Asp216 and Asp220 would still affect the pentamer formation of the full-length MRS2, which would affect the negative feedback mechanism of MRS2. In fact, the authors observed the upregulated Mg²⁺ transport in D216/D220A mutant, but this can be explained by the destabilization of pentamer formation of the full-length MRS2. The authors should revise the interpretation of this mutant in the revised manuscript if they plan to include the data on this mutant in the revised manuscript.

The AlphaFold2 model does not show an interface between Asp216/Asp220 and His232, Gln236 and Lys239. An objective PDBsum analysis (Laskowski *et al*, 2018) of the homopentamer model generated using AlphaFold-Multimer (Evans *et al*, 2022) does not identify any H-bonds, salt bridges or other non-bonded contacts between these residues highlighted by the reviewer. Additionally, our experimental SEC-MALS and DLS data, confirm that the D216A/D220A mutations do not disrupt the stoichiometry of the MRS2 matrix domain or the higher-order assembly of the MRS2 channel (see Fig. 2A, Fig. 3A, Fig. 7, Table S1 and Table S2). Nevertheless, we used the PDBsum analysis of the AlphaFold2 homopentamer to present a possible mechanism for the Asp216/Asp220-dependent, Mg²⁺-binding induced MRS2-NTD disassembly we observed.

Specifically, we mention that Pro221, immediately downstream of Asp220 (*i.e.* DALVDP), H-bonds with Arg228 of an adjacent protomer, and Mg²⁺-binding induced increased α -helicity in the DALVD region (observed in our experimental data) could not only rearrange the loop containing the Pro221 but also the position of the connected helix that mediates many more interprotomer contacts, leading to subunit dissociation (see Discussion, lines 394-404).

References

- Aden E, Carlsson M, Poortvliet E, Stenlund H, Linder J, Edstrom M, Forsgren L, Haglin L (2011) Dietary intake and olfactory function in patients with newly diagnosed Parkinson's disease: a case-control study. *Nutr Neurosci* 14: 25-31
- Barbiroli B, Martinelli P, Patuelli A, Lodi R, Iotti S, Cortelli P, Montagna P (1999) Phosphorus magnetic resonance spectroscopy in multiple system atrophy and Parkinson's disease. *Mov Disord* 14: 430-435
- Bauer PJ (2001) The local Ca concentration profile in the vicinity of a Ca channel. *Cell Biochem Biophys* 35: 49-61
- Chad JE, Eckert R (1984) Calcium domains associated with individual channels can account for anomalous voltage relations of CA-dependent responses. *Biophys J* 45: 993-999
- Daw CC, Ramachandran K, Enslow BT, Maity S, Bursic B, Novello MJ, Rubannelsonkumar CS, Mashal AH, Ravichandran J, Bakewell TM *et al* (2020) Lactate Elicits ER-Mitochondrial Mg(2+) Dynamics to Integrate Cellular Metabolism. *Cell* 183: 474-489 e417
- Evans R, O'Neill M, Pritzel A, Antropova N, Senior A, Green T, Žídek A, Bates R, Blackwell S, Yim J *et al* (2022) Protein complex prediction with AlphaFold-Multimer. *bioRxiv*: 2021.2010.2004.463034
- Fernandez-Sanz C, De la Fuente S, Sheu SS (2019) Mitochondrial Ca(2+) concentrations in live cells: quantification methods and discrepancies. *FEBS Lett* 593: 1528-1541
- Kolisek M, Sponder G, Mastrototaro L, Smorodchenko A, Launay P, Vormann J, Schweigel-Rontgen M (2013) Substitution p.A350V in Na(+)/Mg(2+) exchanger SLC4A1, potentially associated with Parkinson's disease, is a gain-of-function mutation. *PLoS One* 8: e71096
- Laskowski RA, Jablonska J, Pravda L, Varekova RS, Thornton JM (2018) PDBsum: Structural summaries of PDB entries. *Protein Sci* 27: 129-134
- Mallilankaraman K, Doonan P, Cardenas C, Chandramoorthy HC, Muller M, Miller R, Hoffman NE, Gandhirajan RK, Molgo J, Birnbaum MJ *et al* (2012) MICU1 is an essential gatekeeper for MCU-mediated mitochondrial Ca(2+) uptake that regulates cell survival. *Cell* 151: 630-644

Notredame C, Higgins DG, Heringa J (2000) T-Coffee: A novel method for fast and accurate multiple sequence alignment. *J Mol Biol* 302: 205-217

Oyanagi K, Kawakami E, Kikuchi-Horie K, Ohara K, Ogata K, Takahama S, Wada M, Kihira T, Yasui M (2006) Magnesium deficiency over generations in rats with special references to the pathogenesis of the Parkinsonism-dementia complex and amyotrophic lateral sclerosis of Guam. *Neuropathology* 26: 115-128

Schaffers OJM, Hoenderop JGJ, Bindels RJM, de Baaij JHF (2018) The rise and fall of novel renal magnesium transporters. *Am J Physiol Renal Physiol* 314: F1027-F1033

Schlingmann KP, Weber S, Peters M, Niemann Nejsum L, Vitzthum H, Klingel K, Kratz M, Haddad E, Ristoff E, Dinour D *et al* (2002) Hypomagnesemia with secondary hypocalcemia is caused by mutations in TRPM6, a new member of the TRPM gene family. *Nat Genet* 31: 166-170

Sievers F, Wilm A, Dineen D, Gibson TJ, Karplus K, Li W, Lopez R, McWilliam H, Remmert M, Soding J *et al* (2011) Fast, scalable generation of high-quality protein multiple sequence alignments using Clustal Omega. *Mol Syst Biol* 7: 539

Zhang Y, Skolnick J (2005) TM-align: a protein structure alignment algorithm based on the TM-score. *Nucleic Acids Res* 33: 2302-2309

January 19, 2023

RE: Life Science Alliance Manuscript #LSA-2022-01742-TR

Peter B. Stathopoulos
Western University
Department of Physiology and Pharmacology, Schulich School of Medicine and Dentistry
Canada

Dear Dr. Stathopoulos,

Thank you for submitting your revised manuscript entitled "The human MRS2 magnesium binding domain is a regulatory feedback switch for channel activity.". We would be happy to publish your paper in Life Science Alliance pending final revisions necessary to meet our formatting guidelines.

- please address the remaining Reviewer #3 concerns - no additional experimentation is requested at this stage. Please provide along a point by point rebuttal letter listing the revisions made
- please add ORCID ID for the corresponding author--you should have received instructions on how to do so
- please upload your Tables in editable .doc or excel format
- there is a callout for figure 11G and missing the callout for figure 11D - please correct this
- please add a callout for Figure 5C to your main manuscript text
- there is a separate Data Availability section, but looks incomplete (pg. 27)
- the blue line in graphic S3D isn't continuous. Please correct.
- please add scale bars for figure S7

NOTES:

-AU contribution section is provided but labeled as "CRediT AUTHOR STATEMENT." Please label this section as "Author contributions"

A. FINAL FILES:

B. MANUSCRIPT ORGANIZATION AND FORMATTING:

Sincerely,

Reviewer #2 (Comments to the Authors (Required)):

Response reviewer #2:

The authors have adequately addressed the points raised by me. My points raised on the relevance of Ca²⁺ in MRS2 regulation and the role of the double mutant MRS2 in mitochondrial Mg²⁺ control have been integrated in the manuscript. In addition, the manuscript has been revised to include my concerns on the MagGreen and microscopy experiments. With this, I think that the manuscript is suitable for publication in the journal and will provide insights relevant for the field.

Reviewer #3 (Comments to the Authors (Required)):

As I mentioned in my last review comments, this work by the group of Dr. Stathopoulos characterized human MRS2 Mg²⁺ transporter by a combination of biochemistry and cell biology-based assays, and would attract a wide range of interests for readers in the field. However, multiple problems still remain in the revised manuscript. Thus, before publication, the authors should extensively revise the manuscript based on my comments below.

1. "Our sequence alignment in Fig. 1B was performed using T-COFFEE [(Notredame et al., 2000); 8,100 citations in Google Scholar]. Addition of 13 more bacterial CorA sequences does not change the alignment of the human DALVD sequence with the identical DALVD sequence of *T. maritima* using T-COFFEE. In contrast, using Clustal Omega [(Sievers et al., 2011); 13,100 citations in Google Scholar], the human DALVD sequence aligns with a different part of the *T. maritima* sequence, as the reviewer points out. Note that the 13 bacterial species included are the complete list curated in the NCBI Landmark Database spanning a diverse, non-redundant and wide taxonomic range.

Thus, as suggested, we have added the multiple sequence alignments output by T-COFFEE and Clustal Omega that include more bacterial CorA sequences, highlighting the algorithm-dependent differences in alignment due to the poor sequence conservation between human MRS2 and bacterial CorA. (see Results lines 201-205, Fig. 1B legend lines 822-825 and new Fig. S1 and Fig. S2)."

We understand the efforts by the authors, but unfortunately, they showed only the sequence alignment generated by T-COFFEE

in the main figure and used it for most of discussion in the revised manuscript, in particular for one of their main conclusions that the Mg²⁺ binding site of hMRS2 is located at the DALVD motif.

There is no reason to justify prioritizing T-COFFEE, that was developed in 2000, over Clustal Omega, which is developed more recently and more commonly used nowadays. The authors should not do such "cherry picking" to justify their statements. The AlphaFold2 model in the pentameric form (See the comment 2 below) is also more consistent with the sequence alignment by Clustal Omega.

2. "Our original manuscript showed the AlphaFold2 predicted structure of human MRS2 in Fig. 10B, downloaded directly from the AlphaFold Protein Structure Database (<https://alphafold.ebi.ac.uk/entry/Q9HD23>). We also generated our own human MRS2 model using a local implementation of AlphaFold2 (v2.2), which appears identical to the downloaded prediction through the structured regions. If the reviewer has generated a model that "looked more reasonable" than the one shown in our original Fig. 10B (Fig. 11B in the revised submission), there may be an error in their implementation or usage of AlphaFold2, as we feel it is unlikely that the deposited MRS2 model and a similar but independently generated AlphaFold2 model are both different than the version generated by the reviewer."

There may be some misunderstanding by the authors. The new AlphaFold2 model in the pentameric form, which is provided by the authors, is now consistent with what I meant in my last review comments. I asked the authors to update the model because the authors initially used the homology model based on TmCorA or the monomeric form of the AlphaFold2.

3. "Indeed, we showed the AlphaFold2 structure in our original Fig. 10B, which must have been overlooked by the reviewer. We note that the AlphaFold2 prediction confidence for the Asp216 and Asp220 residues (the focus of our experimental work) of human MRS2 is not considered very high (i.e. pLDDT < 90). Additionally, the functional stoichiometry of the human MRS2 channel remains unknown, although we agree that a homopentamer is probable. Nevertheless, we also generated a pentameric structure using AlphaFold2, as suggested. Just as for the monomer prediction, the Asp216 and Asp220 residues and entire DALVD sequence show pLDDT scores of < 90. Therefore, while the backbone conformations predicted for these residues are probably correct, the predicted orientations of the side chains are not reliable, and an experimentally determined human MRS2 structure is needed to firmly establish the precise functional stoichiometry of the channel and the positions of the Asp216 and Asp220 side chains with respect to the assembled channel.

As suggested, the AlphaFold2 pentameric model is now shown as new Fig. 11B and new Fig. 11C. Additionally, we have added text in the Fig. 11 legend, lines 1009-1014, describing the AlphaFold2 prediction confidence for DALVD and adjacent loop sequence and reiterating that an experimentally determined human MRS2 high-resolution structure is needed to firmly establish the functional stoichiometry of the channel and the positions of the Asp216 and Asp220 side chains with respect to the assembled channel. Also, given the concerns regarding the alignment, we have eliminated the Modeller model from the manuscript and solely focused on the AlphaFold2 prediction."

As shown in the AlphaFold2 model by the authors in Figure 11, the location of the DALVD motif in TmCorA and hMRS2 are totally different, which is also not consistent with the T-COFFEE-based sequence alignment. If the T-COFFEE-based sequence alignment is right, these should be located in a similar position. Instead, the sequence alignment by Clustal Omega is consistent with the AlphaFold2 model in Figure 11. In other words, if they use the AlphaFold 2 model, they should not use the T-COFFEE based sequence alignment and should use Clustal Omega based sequence alignment. In this case, the DALVD motif does not exist.

I understand that they wish to use the T-COFFEE based sequence alignment to support their statement that DALVD motif form a Mg²⁺ binding site. However, as I mentioned above, such motif does not exist in the sequence alignment by ClustalOmega. So, please delete all statements regarding the DALVD motif, because it does not exist as a conserved motif. They can mention Asp216 and Asp220 residues based on the experiment but not as the DALVD motif.

4. "We thank the reviewer for suggesting we look closer at the alignment. Since the T-COFFEE and Clustal Omega output different alignments (see our response to reviewer #3, point #1 above), we have revised the last sentence to read "Remarkably, human MRS2 contains an identical DALVD sequence, which could similarly coordinate Mg²⁺" (see Results, lines 200-201)."

This is not enough. As I mentioned above, please delete all description regarding the DALVD motif. Either Clustal Omega or AlphaFold2 does not support the existence of such motif.

5. The suggestion that we "should not mention Asp216 and Asp220 as a Mg²⁺ binding site" is unjustified as it overlooks the experimental data we collected showing Asp216/Asp220-dependent Mg²⁺ binding and structural sensitivity (i.e. see Fig. 5, Fig. 6, Fig. 7, Fig. 8, Fig. S3, Fig. S4, Table S1, Table S2 and Table S3). These experimental observations are the focus of the manuscript and are independent of the alignment and model predictions focused on by this reviewer. We stress that the models generated in Fig. 11 are only predictions and discussion points used to speculate how Mg²⁺ binding at the Asp216/Asp220 site could result channel inhibition.

Our data showed that Asp216Ala/Asp220Ala mutations disrupt Mg²⁺-binding (see Fig. 5A and Fig. 5D; Table S3) and the associated biophysical and structural effects without altering the protein secondary structure (see Fig 8A and 8B), stability (see Fig. 8E and Fig. 8F), tertiary structure (see Fig 6), quaternary structures and higher order assemblies (see Fig. 2A versus Fig. 7A and Fig. 7B, Fig. 3A versus Fig. 7C, Table S1 and Table S2). Nevertheless, we thank the reviewer for pointing out this

additional divalent cation binding site possibility. Thus, we have indicated that Glu243, Asp247, Asp305 and Glu312 may be involved in forming additional cation binding site(s), besides the binding site we have experimentally identified and validated (see new Fig. 11C, Discussion, lines 404-408)."

Thank you for mentioning Glu243, Asp247, Asp305 and Glu312 in the revised manuscript.

6. "The AlphaFold2 model does not show an interface between Asp216/Asp220 and His232, Gln236 and Lys239. An objective PDBsum analysis (Laskowski et al, 2018) of the homopentamer model generated using AlphaFold-Multimer (Evans et al, 2022) does not identify any H-bonds, salt bridges or other non-bonded contacts between these residues highlighted by the reviewer. Additionally, our experimental SEC-MALS and DLS data, confirm that the D216A/D220A mutations do not disrupt the stoichiometry of the MRS2 matrix domain or the higher-order assembly of the MRS2 channel (see Fig. 2A, Fig. 3A, Fig. 7, Table S1 and Table S2). Nevertheless, we used the PDBsum analysis of the AlphaFold2 homopentamer to present a possible mechanism for the Asp216/Asp220-dependent, Mg²⁺-binding induced MRS2-NTD disassembly we observed. Specifically, we mention that Pro221, immediately downstream of Asp220 (i.e. DALVDP), H-bonds with Arg228 of an adjacent protomer, and Mg²⁺-binding induced increased α -helicity in the DALVD region (observed in our experimental data) could not only rearrange the loop containing the Pro221 but also the position of the connected helix that mediates many more interprotomer contacts, leading to subunit dissociation (see Discussion, lines 394-404)."

Please check the figure I generated from AlphaFold2 model (<https://ufile.io/20j55yhn>). Asp216 and Asp220 are located at the subunit interface near His232, Gln236 and Lys239. So, there is another possibility for the interpretation of D216A/D220A mutant, as I mentioned in my last review comments. The authors should point it out in the revised manuscript. I hope the authors understand that I am writing review comments to help the authors. I basically support this manuscript for publication as long as the manuscript is properly revised.

Our responses (**red text**) to the reviewer #3 comments (*italics*) are shown below.

As I mentioned in my last review comments, this work by the group of Dr. Stathopoulos characterized human MRS2 Mg²⁺ transporter by a combination of biochemistry and cell biology-based assays, and would attract a wide range of interests for readers in the field. However, multiple problems still remain in the revised manuscript. Thus, before publication, the authors should extensively revise the manuscript based on my comments below.

We thank the reviewer for reinforcing the importance of the work.

1. We understand the efforts by the authors, but unfortunately, they showed only the sequence alignment generated by T-COFFEE in the main figure and used it for most of discussion in the revised manuscript, in particular for one of their main conclusions that the Mg²⁺ binding site of hMRS2 is located at the DALVD motif.

There is no reason to justify prioritizing T-COFFEE, that was developed in 2000, over Clustal Omega, which is developed more recently and more commonly used nowadays. The authors should not do such "cherry picking" to justify their statements. The AlphaFold2 model in the pentameric form (See the comment 2 below) is also more consistent with the sequence alignment by Clustal Omega.

To prevent any perceived bias and readers from misinterpreting our statements regarding the DALVD region, we have now replaced the T-COFFEE alignment with the Clustal Omega alignment in the main figure (Fig. 1B) (see new Fig. 1B).

We also note here that since 2022, the primary T-COFFEE and Clustal Omega publications have been cited 366 and 1730 times, respectively (Google Scholar), suggesting both programs remain widely used. Additionally, the only statement we make in the main-text regarding the T-COFFEE alignments is "However, sequence-based alignment of the bacterial and vertebrate DALVD regions are algorithm-dependent due to the poor sequence conservation: T-COFFEE (Notredame *et al.*, 2000) aligns the human and *T. maritima* DALVD stretches as conserved whereas Clustal Omega (Sievers *et al.*, 2011) does not (Fig. S1 and S2)." Since we make no reference to the T-COFFEE alignment after this description, we disagree with the reviewer that we are using the T-COFFEE alignment for most of the Discussion (*i.e.* we make no other statements regarding the T-COFFEE alignment).

2. There may be some misunderstanding by the authors. The new AlphaFold2 model in the pentameric form, which is provided by the authors, is now consistent with what I meant in my last review comments. I asked the authors to update the model because the authors initially used the homology model based on TmCorA or the monomeric form of the AlphaFold2.

We thank the reviewer for confirmation.

3. As shown in the AlphaFold2 model by the authors in Figure 11, the location of the DALVD motif in TmCorA and hMRS2 are totally different, which is also not consistent with the T-COFFEE-based sequence alignment. If the T-COFFEE-based sequence alignment is right, these

should be located in a similar position. Instead, the sequence alignment by Clustal Omega is consistent with the AlphaFold2 model in Figure 11. In other words, if they use the AlphaFold 2 model, they should not use the T-COFFEE based sequence alignment and should use Clustal Omega based sequence alignment. In this case, the DALVD motif does not exist.

I understand that they wish to use the T-COFFEE based sequence alignment to support their statement that DALVD motif form a Mg²⁺ binding site. However, as I mentioned above, such motif does not exist in the sequence alignment by ClustalOmega. So, please delete all statements regarding the DALVD motif, because it does not exist as a conserved motif. They can mention Asp216 and Asp220 residues based on the experiment but not as the DALVD motif.

We do not use the T-COFFEE-based sequence alignment to support our statement that DALVD forms a Mg²⁺ binding site. Our rationale is simply that human MRS2 and *T. maritima* CorA both contain a DALVD sequence stretch in homologous helices and domains. In the case of *T. maritima* CorA, the Asp residues of DALVD coordinate Mg²⁺. Thus, we experimentally assessed whether the Asp residues of human DALVD also coordinate Mg²⁺. This rationale is independent of the multiple sequence alignments shown in **Fig. 1**, **Fig. S1** and **Fig. S2**.

We use the term ‘DALVD’ in the manuscript to provide context on the relative position of the two Asp residues and the intervening residue composition. The term not only shows readers that the two Asp residues are at positions *i* and *i*+4 of this sequence stretch but also that there are no helix-breaking residues situated between them, which would orient both Asp side chains on the same side of an α -helix and adjacent in 3D space (even though they are not adjacent in sequence space) and permit both to interact with the same Mg²⁺ (**see new lines 209-213**).

We do not describe DALVD as a ‘**conserved motif**’ or ‘**motif**’ in the manuscript. We only refer to DALVD as a sequence or region since residues 216-220 of human MRS2 exist as Asp-Ala-Leu-Val-Asp (DALVD) (see NCBI Accession #NP_065713.1). To avoid this misinterpretation by other readers, we now explicitly state “Our use of the term DALVD in this study is in reference to the stretch of polypeptide chain containing the D216 and D220 residues and does not imply a motif that is conserved with the DALVD stretch of *T. maritima* CorA” (**see new lines 207-213**). We have also deleted nine instances of the DALVD term in the Discussion to reduce the emphasis on this specific sequence, as detailed in the point below.

4. This is not enough. As I mentioned above, please delete all description regarding the DALVD motif. Either Clustal Omega or AlphaFold2 does not support the existence of such motif.

As mentioned above, we do not refer to DALVD as a ‘**motif**’, ‘**conserved motif**’ or ‘**DALVD motif**’ in the manuscript. To reinforce that this is not a strictly conserved motif, we have now additionally stated that the AlphaFold2 model agrees with the non-conserved Clustal Omega positioning of bacterial and vertebrate DALVD regions (**see new lines 205-207**). Additionally, to prevent this misunderstanding, we have now deleted nine instances of ‘DALVD’ relating to human MRS2 (**see edited lines 217, 394, 401, 405, 407, 408, 409, 410 and 846**). Along with deleting these DALVD terms and adding a statement that our use of the term DALVD does not imply a conserved motif with bacterial CorA (see above point), we have also removed the word “identical” used on two occasions to prevent misinterpretation as sequence identity (**see edited**

lines 200 and 204). Since reviewer #2 asked for a comparison between yeast and human DALVD sequences and we show multiple sequence as well as structural alignments comparing the locations of the DALVD sequences, we hope the reviewer understands that we are unable to remove all our descriptions regarding DALVD.

5. Thank you for mentioning Glu243, Asp247, Asp305 and Glu312 in the revised manuscript.

We thank the reviewer for confirmation.

6. Please check the figure I generated from AlphaFold2 model (<https://ufile.io/20j55yhn>). Asp216 and Asp220 are located at the subunit interface near His232, Gln236 and Lys239. So, there is another possibility for the interpretation of D216A/D220A mutant, as I mentioned in my last review comments. The authors should point it out in the revised manuscript. I hope the authors understand that I am writing review comments to help the authors. I basically support this manuscript for publication as long as the manuscript is properly revised.

We have added text pointing out that since D216 and D220 are predicted to be near H232, Q236 and K239 of an adjacent protomer, mutations to the Asp residues could potentially perturb inter-protomer interactions involving these residues. In this scenario, the destabilization of inter-domain interactions would lead to increased open probability, similar to the Mg²⁺-dissociation dependent mechanism for *T. maritima* CorA, recently articulated in detail by the A. Guskov group (PMID 33886304). However, this scenario appears to be inconsistent with the Mg²⁺-binding induced dissociation we observed for the human MRS2 matrix domain and our observations that the D216A/D220A mutation does not alter the stoichiometry of the human MRS2 matrix domain or the full-length human MRS2 assembly in CHAPS micelles (**see lines 477-485 and modified Fig. 11C**).

January 26, 2023

RE: Life Science Alliance Manuscript #LSA-2022-01742-TRR

Peter B. Stathopoulos
Western University
Physiology and Pharmacology, Schulich School of Medicine and Dentistry
1151 Richmond Street
London, Ontario N6A5C1
Canada

Dear Dr. Stathopoulos,

Thank you for submitting your Research Article entitled "The human MRS2 magnesium binding domain is a regulatory feedback switch for channel activity.". It is a pleasure to let you know that your manuscript is now accepted for publication in Life Science Alliance. Congratulations on this interesting work.

DISTRIBUTION OF MATERIALS:

Again, congratulations on a very nice paper. I hope you found the review process to be constructive and are pleased with how the manuscript was handled editorially. We look forward to future exciting submissions from your lab.

Sincerely,
